# LAST-ITERATE CONVERGENCE PROPERTIES OF REGRET-MATCHING ALGORITHMS IN GAMES

## ABSTRACT

Algorithms based on regret matching, specifically regret matching$^+$ (RM$^+$), and its variants are the most popular approaches for solving large-scale two-player zero-sum games in practice. Unlike algorithms such as optimistic gradient descent ascent, which have strong last-iterate and ergodic convergence properties for zero-sum games, virtually nothing is known about the last-iterate properties of regret-matching algorithms. Given the importance of last-iterate convergence for numerical optimization reasons and relevance as modeling real-word learning in games, in this paper, we study the last-iterate convergence properties of various popular variants of RM$^+$. First, we show numerically that several practical variants such as simultaneous RM$^+$, alternating RM$^+$, and simultaneous predictive RM$^+$, all lack last-iterate convergence guarantees even on a simple $3 \times 3$ game. We then prove that recent variants of these algorithms based on a *smoothing* technique do enjoy last-iterate convergence: we prove that *extragradient RM$^+$* and *smooth Predictive RM$^+$* enjoy asymptotic last-iterate convergence (without a rate) and $1/\sqrt{t}$ best-iterate convergence. Finally, we introduce restarted variants of these algorithms, and show that they enjoy linear-rate last-iterate convergence.

## 1 INTRODUCTION

Saddle-point optimization problems have attracted significant research interest with applications in generative adversarial networks (Goodfellow et al., 2020), imaging (Chambolle & Pock, 2011), market equilibrium (Kroer et al., 2019), and game-solving (Von Stengel, 1996). Matrix games provide an elementary saddle-point optimization setting, where the set of decisions of each player is a simplex and the objective function is bilinear. Matrix games can be solved via self-play, where each player employs a *regret minimizer*, such as online gradient descent ascent (GDA), multiplicative weight updates (MWU), or Regret Matching$^+$ (RM$^+$). In this case, a well-known folk theorem shows that the *average* of the strategies visited at all iterations converges to a Nash equilibrium of the matrix game, at a rate of $O(1/\sqrt{T})$ for GDA, MWU and RM$^+$, and at a rate of $O(1/T)$ for predictive variants of GDA and MWU (Syrgkanis et al., 2015; Rakhlin & Sridharan, 2013).

In recent years, there has been increasing interest in the *last-iterate* convergence properties of algorithms for saddle-point problems (Daskalakis & Panageas, 2019; Golowich et al., 2020; Wei et al., 2021; Lee et al., 2021). There are multiple reasons for this. First, since no-regret learning is often viewed as a plausible method of real-world gameplay, it would be desirable to have the actual strategy iterates to converge to an equilibrium instead of only having the average converge. Secondly, suppose self-play via no-regret learning is being used to compute an equilibrium. In that case, iterate averaging can often be cumbersome, especially when deep-learning components are involved in the learning approach, since it may not be possible to average the outputs of a neural network. Thirdly, iterate averaging may be slower to converge since the convergence rate is limited by the extent to which early "bad" iterates are discounted in the average. Even in the simple matrix game setting, interesting questions arise when considering the last-iterate convergence properties of widely used algorithms. For instance, both GDA and MWU may diverge (Bailey & Piliouras, 2018; Cheung & Piliouras, 2019), whereas their predictive counterparts, Optimistic GDA (OGDA) (Daskalakis et al., 2018; Mertikopoulos et al., 2019; Wei et al., 2021) and Optimistic MWU (OMWU) (Daskalakis & Panageas, 2019; Lei et al., 2021) converge at a linear rate under some assumptions on the matrix

games. Furthermore, it has been demonstrated that OGDA has a last-iterate convergence rate of $O(1/\sqrt{T})$ without any assumptions on the game (Cai et al., 2022; Gorbunov et al., 2022b).

On the other hand, very little is known about the last-iterate convergence properties of RM$^+$ and its variants, despite their huge popularity in solving large-scale games in practice. Specifically, RM$^+$ (Hart & Mas-Colell, 2000; Tammelin et al., 2015) is a simple, stepsize-free regret minimizer, guaranteeing $O(1/\sqrt{T})$ ergodic convergence in self-play for matrix games. When combined with counterfactual regret minimization (Zinkevich et al., 2007), linear averaging, and alternation, RM$^+$ has been used in multiple important milestones in solving extremely large poker games (Bowling et al., 2015; Moravčík et al., 2017; Brown & Sandholm, 2018). Despite its very strong empirical performance, the last-iterate behavior of (vanilla) RM$^+$ may diverge even on small matrix games, e.g., rock-paper-scissors (Lee et al., 2021).

Moreover, unlike OGDA and OMWU, the predictive variants of RM$^+$ do not enjoy similar theoretical speed up. Specifically, predictive RM$^+$ (PRM$^+$), a predictive variant of RM$^+$, was introduced in (Farina et al., 2021) and shown to achieve very good empirical performance in some games. However, it still only guarantees $O(1/\sqrt{T})$ ergodic convergence for matrix games (unlike OGDA and OMWU). To address this, Farina et al. (2023) introduce two variants of RM$^+$ with $O(1/T)$ ergodic convergence for matrix games, namely, Extragradient RM$^+$ (ExRM$^+$) and Smooth Predictive RM$^+$ (SPRM$^+$), but the last-iterate convergence properties of all these variants remain unknown.[1]

Motivated by this, in this work, we study both theoretically and empirically the last-iterate behavior of RM$^+$ and its various variants. Our **main results** are as follows.

1. We provide numerical evidence that RM$^+$ and important variants of RM$^+$, including alternating RM$^+$ and PRM$^+$, may fail to have asymptotic last-iterate convergence. Conversely, we also prove that RM$^+$ *does* have last-iterate convergence in a very restrictive setting, where the matrix game admits a strict Nash equilibrium.

2. We then study the convergence properties of two recently proposed variants of RM$^+$: ExRM$^+$ (Algorithm 4) and SPRM$^+$ (Algorithm 5). For these two algorithms, we first prove the asymptotic convergence of the last iterate (without providing a rate). We then show a $O(1/\sqrt{t})$-rate for the duality gap for the best iterate after $t$ iterations. Building on this last observation, we finally introduce new variants of ExRM$^+$ and SPRM$^+$ that restart whenever the distance between two consecutive iterates has been halved, and prove that they enjoy linear last-iterate convergence.

3. Existing last-iterate convergence results rely heavily on the monotonicity assumption (or even strong monotonicity), while RM$^+$ and its variants are equivalent to solving a variational inequality (VI) problem with a *non-monotone* operator which satisfies the Minty condition. Our proofs use new observations on the structure of the solution set and provide characterizations of the limit points of the learning dynamics, which might be of independent interest for analyzing last-iterate convergence of other algorithms.

4. We verify the last-iterate convergence of ExRM$^+$ and SPRM$^+$ (including their restarted variants that we propose) numerically on four instances of matrix games, including Kuhn poker and Goofspiel. We also note that while vanilla RM$^+$, alternating RM$^+$, and SPRM$^+$ may not converge, alternating PRM$^+$ exhibits a surprisingly fast last-iterate convergence.

**Notation.** We write $\mathbf{0}$ for the vector with $0$ on every component and $\mathbf{1}_d$ for the vector in $\mathbb{R}^d$ with $1$ on every component. We use the convention that $\mathbf{0}/0 = (1/d)\mathbf{1}_d$. We denote by $\Delta^d$ the simplex: $\Delta^d = \{x \in \mathbb{R}_+^d \mid \langle x, \mathbf{1}_d \rangle = 1\}$. For $x \in \mathbb{R}$, we write $[x]^+$ for the positive part of $x : [x]^+ = \max\{0, x\}$, and we overload this notation to vectors component-wise. We use $\|z\|$ to denote the $\ell_2$ norm of a vector $z$.

## 2 PRELIMINARIES ON REGRET MATCHING$^+$ AND SELF PLAY

In this paper, we study iterative algorithms for solving the following matrix game:

$$\min_{x \in \Delta^{d_1}} \max_{y \in \Delta^{d_2}} x^\top A y \tag{1}$$

---

[1]We note that a recent work (Meng et al., 2023) does study the last-iterate convergence of RM$^+$ variants, but for strongly-convex strongly-concave games, which is incomparable to our matrix game setting.

for a *payoff matrix* $A \in \mathbb{R}^{d_1 \times d_2}$. We define $\mathcal{Z} = \Delta^{d_1} \times \Delta^{d_2}$ to be the set of feasible pairs of strategies. The duality gap of a pair of feasible strategy $(x, y) \in \mathcal{Z}$ is defined as

$$\text{DualityGap}(x, y) := \max_{y' \in \Delta^{d_2}} x^\top A y' - \min_{x' \in \Delta^{d_1}} x'^\top A y.$$

Note that we always have $\text{DualityGap}(x, y) \geq 0$, and it is well-known that $\text{DualityGap}(x, y) \leq \epsilon$ implies that the pair $(x, y) \in \mathcal{Z}$ is an $\epsilon$-Nash equilibrium of the matrix game (1). When both players of (1) employ a *regret minimizer*, a well-known folk theorem shows that the averages of the iterates generated during self-play converge to a Nash equilibrium of the game (Freund & Schapire, 1999). This framework can be instantiated with any regret minimizers, for instance, online mirror descent, follow-the-regularized leader, regret matching, and optimistic variants of these algorithms. We refer to (Hazan et al., 2016) for an extensive review on regret minimization. From here on, we focus on solving (1) via Regret Matching$^+$ and its variants. To describe these algorithms, it is useful to define for a strategy $x \in \Delta^d$ and a loss vector $\ell \in \mathbb{R}^d$, the negative instantaneous regret vector $f(x, \ell) = \ell - x^\top \ell \cdot \mathbf{1}_d,$[2] and also define the normalization operator $g : \mathbb{R}_+^{d_1} \times \mathbb{R}_+^{d_2} \to \mathcal{Z}$ such that for $z = (z_1, z_2) \in \mathbb{R}_+^{d_1} \times \mathbb{R}_+^{d_2}$, we have $g(z) = (z_1/\|z_1\|_1, z_2/\|z_2\|_1) \in \mathcal{Z}$.

**Regret Matching$^+$ (RM$^+$), alternation, and Predictive RM$^+$.** We describe Regret Matching$^+$ (RM$^+$) in Algorithm 1 (Tammelin, 2014).[3] It maintains two sequences: a sequence of joint *aggregate payoffs* $(R_x^t, R_y^t) \in \mathbb{R}_+^{d_1} \times \mathbb{R}_+^{d_2}$ updated using the instantaneous regret vector, and a sequence of joint strategies $(x^t, y^t) \in \mathcal{Z}$ directly normalized from the aggregate payoff. Note that the update rules are stepsize-free and only perform closed-form operations (thresholding and rescaling).

A popular variant of RM$^+$, *Alternating RM$^+$* (Tammelin et al., 2015; Burch et al., 2019), is shown in Algorithm 2. In alternating RM$^+$, the updates between the two players are asynchronous, and at iteration $t$, the second player observes the choice $x^{t+1}$ of the first player when choosing their own decision $y^{t+1}$. Alternation leads to faster empirical performance for solving matrix and extensive-form games, even though the theoretical guarantees remain the same as for vanilla RM$^+$ (Burch et al., 2019; Grand-Clément & Kroer, 2023).

---

**Algorithm 1** Regret Matching$^+$ (RM$^+$)

1: **Initialize:** $(R_x^0, R_y^0) = \mathbf{0}$, $(x^0, y^0) \in \mathcal{Z}$
2: **for** $t = 0, 1, \ldots$ **do**
3: $\quad R_x^{t+1} = [R_x^t - f(x^t, Ay^t)]^+$
4: $\quad R_y^{t+1} = [R_y^t + f(y^t, A^\top x^t)]^+$
5: $\quad (x^{t+1}, y^{t+1}) = g(R_x^{t+1}, R_y^{t+1})$

**Algorithm 2** Alternating RM$^+$ (alt. RM$^+$)

1: **Initialize:** $(R_x^0, R_y^0) = \mathbf{0}$, $(x^0, y^0) \in \mathcal{Z}$
2: **for** $t = 0, 1, \ldots$ **do**
3: $\quad R_x^{t+1} = [R_x^t - f(x^t, Ay^t)]^+$
4: $\quad x^{t+1} = \dfrac{R_x^{t+1}}{\|R_x^{t+1}\|_1}$
5: $\quad R_y^{t+1} = [R_y^t + f(y^t, A^\top x^{t+1})]^+$
6: $\quad y^{t+1} = \dfrac{R_y^{t+1}}{\|R_y^{t+1}\|_1}$

---

Finally, we describe Predictive RM$^+$ (PRM$^+$) from (Farina et al., 2021) in Algorithm 3. PRM$^+$ incorporates *predictions* of the next losses faced by each player (using the most recent observed losses) when computing the strategies at each iteration, akin to predictive/optimistic online mirror descent (Rakhlin & Sridharan, 2013; Syrgkanis et al., 2015).

In practice, PRM$^+$ can also be combined with alternation, but despite

---

**Algorithm 3** Predictive RM$^+$ (PRM$^+$)

1: **Initialize:** $(R_x^0, R_y^0) = \mathbf{0}$, $(x_0, y_0) \in \mathcal{Z}$
2: **for** $t = 0, 1, \ldots$ **do**
3: $\quad R_x^{t+1} = [R_x^t - f(x^t, Ay^t)]^+$
4: $\quad R_y^{t+1} = [R_y^t + f(y^t, A^\top x^t)]^+$
5: $\quad (x^{t+1}, y^{t+1}) =$
$\quad\quad g([R_x^{t+1} - f(x^t, Ay^t)]^+, [R_y^{t+1} + f(y^t, A^\top x^t)]^+)$

---

[2]Here, $d$ can be either $d_1$ or $d_2$. That is, we overload the notation $f$ so its domain depends on the inputs.

[3]Typically, RM$^+$ and PRM$^+$ are introduced as regret minimizers that return a sequence of decisions against any sequence of losses (Tammelin et al., 2015; Farina et al., 2021). For conciseness, we directly present them as self-play algorithms for solving matrix games, as in Algorithm 1 and Algorithm 3.

strong empirical performance, it is unknown if alternating PRM$^+$ enjoys ergodic convergence. In contrast, based on the aforementioned folk theorem and the regret guarantees of RM$^+$ (Tammelin, 2014), alternating RM$^+$ (Burch et al., 2019), and PRM$^+$ (Farina et al., 2021), the duality gap of the average strategy of all these algorithms goes down at a rate of $O(1/\sqrt{T})$. However, we will show in the next section that the iterates $(x^t, y^t)$ themselves may not converge. We also note that despite the connections between PRM$^+$ and predictive online mirror descent, PRM$^+$ does not achieve $O(1/T)$ ergodic convergence, because of its lack of stability (Farina et al., 2023).

**Extragradient RM$^+$ and Smooth Predictive RM$^+$**   We now describe two theoretically-faster variants of RM$^+$ recently introduced in (Farina et al., 2023). To provide a concise formulation, we first need some additional notation. First, we define the clipped positive orthant $\Delta^{d_i}_{\geq} := \{u \in \mathbb{R}^{d_i}_+ : u^\top \mathbf{1}_{d_i} \geq 1\}$ for $i = 1, 2$ and $\mathcal{Z}_{\geq} = \Delta^{d_1}_{\geq} \times \Delta^{d_2}_{\geq}$. For a point $z \in \mathcal{Z}_{\geq}$, we often write it as $z = (Rx, Qy)$ for positive real numbers $R$ and $Q$ such that $(x, y) = g(z)$. Moreover, we define the operator $F : \mathcal{Z}_{\geq} \to \mathbb{R}^{d_1+d_2}$ as follows: for any $z = (Rx, Qy) \in \mathcal{Z}_{\geq}$,

$$F(z) = F((Rx, Qy)) = \begin{bmatrix} f(x, Ay) \\ f(y, -A^\top x) \end{bmatrix} = \begin{bmatrix} Ay - x^\top Ay \cdot \mathbf{1}_{d_1} \\ -A^\top x + x^\top Ay \cdot \mathbf{1}_{d_2} \end{bmatrix}, \tag{2}$$

which is $L_F$-Lipschitz-continuous over $\mathcal{Z}_{\geq}$ with $L_F = \sqrt{6}\|A\|_{op} \max\{d_1, d_2\}$ (Farina et al., 2023). We also write $\Pi_{\mathcal{Z}_{\geq}}(u)$ for the $L_2$ projection onto $\mathcal{Z}_{\geq}$ of the vector $u$: $\Pi_{\mathcal{Z}_{\geq}}(u) = \arg\min_{z' \in \mathcal{Z}_{\geq}} \|z' - u\|_2$. With these notations, Extragradient RM$^+$ (ExRM$^+$) and Smooth PRM$^+$ (SPRM$^+$) are defined in Algorithm 4 and in Algorithm 5.

| **Algorithm 4** Extragradient RM$^+$ (ExRM$^+$) | **Algorithm 5** Smooth PRM$^+$ (SPRM$^+$) |
|---|---|
| 1: **Input**: Step size $\eta \in (0, \frac{1}{L_F})$. | 1: **Input**: Step size $\eta \in (0, \frac{1}{8L_F}]$. |
| 2: **Initialize**: $z^0 \in \mathcal{Z}$ | 2: **Initialize**: $z^{-1} = w^0 \in \mathcal{Z}$ |
| 3: **for** $t = 0, 1, \ldots$ **do** | 3: **for** $t = 0, 1, \ldots$ **do** |
| 4: $\quad z^{t+1/2} = \Pi_{\mathcal{Z}_{\geq}}(z^t - \eta F(z^t))$ | 4: $\quad z^t = \Pi_{\mathcal{Z}_{\geq}}(w^t - \eta F(z^{t-1}))$ |
| 5: $\quad z^{t+1} = \Pi_{\mathcal{Z}_{\geq}}(z^t - \eta F(z^{t+1/2}))$ | 5: $\quad w^{t+1} = \Pi_{\mathcal{Z}_{\geq}}(w^t - \eta F(z^t))$ |

ExRM$^+$ is connected to the Extragradient (EG) algorithm (Korpelevich, 1976) and SPRM$^+$ is connected to the Optimistic Gradient algorithm (Popov, 1980; Rakhlin & Sridharan, 2013) (see Section 4 and Section 5 for details). Farina et al. (2023) show that ExRM$^+$ and SPRM$^+$ enjoy fast $O(\frac{1}{T})$ ergodic convergence for solving matrix games.

## 3   NON-CONVERGENCE OF RM$^+$, ALTERNATING RM$^+$, AND PRM$^+$

In this section, we show empirically that several existing variants of RM$^+$ may not converge in iterates. Specifically, we numerically investigate four algorithms—RM$^+$, alternating RM$^+$, PRM$^+$, and alternating PRM$^+$—on a simple $3 \times 3$ game matrix $A = [[3, 0, -3], [0, 3, -4], [0, 0, 1]]$ that has the unique Nash equilibrium $(x^\star, y^\star) = ([\frac{1}{12}, \frac{1}{12}, \frac{5}{6}], [\frac{1}{3}, \frac{5}{12}, \frac{1}{4}])$. The same instance was also used in (Farina et al., 2023) to illustrate the instability of PRM$^+$ and slow ergodic convergence of RM$^+$ and PRM$^+$. The results are shown in Figure 1. We observe that for RM$^+$, alternating RM$^+$, and PRM$^+$, the duality gap remains on the order of $10^{-1}$ even after $10^5$ iterations. Our empirical findings are in line with Theorem 3 of Lee et al. (2021), who pointed out that RM$^+$ diverges on the rock-paper-scissors game. In contrast, alternating PRM$^+$ enjoys good last-iterate convergence properties on this matrix game instance. Overall, our empirical results suggest that RM$^+$, alternating RM$^+$, and PRM$^+$ all fail to converge in iterates, *even when the game has a unique Nash equilibrium*, a more regular and benign setting than the general case.

We complement our empirical non-convergence results by showing that RM$^+$ has asymptotic convergence under the restrictive assumption that the game has a *strict* Nash equilibrium. To our knowledge, this is the first positive last-iterate convergence result related to RM$^+$. In a strict Nash equilibrium $(x^\star, y^\star)$, $x^\star$ is the *unique* best-response to $y^\star$ and *vice versa*. It follows from this definition that the equilibrium is unique and that $x^\star, y^\star$ are pure strategies. As an example, the game matrix $A = [[2, 1], [3, 4]]$ has a strict Nash equilibrium $x^\star = [1, 0]$ and $y^\star = [1, 0]$.

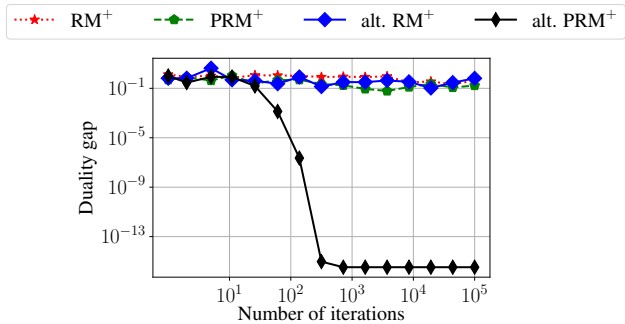

Figure 1: Duality gap of the current iterates generated by $RM^+$, $RM^+$ with alternation, Predictive $RM^+$ ($PRM^+$), and Predictive $RM^+$ with alternation on the zero-sum game with payoff matrix $A = [[3, 0, -3], [0, 3, -4], [0, 0, 1]]$.

**Theorem 1** (Convergence of $RM^+$ to Strict NE). *If a matrix game has a* strict *Nash equilibrium* $(x^\star, y^\star)$, $RM^+$ *(Algorithm 1) converges in last-iterate, that is,* $\lim_{t \to \infty}\{(x^t, y^t)\} = (x^\star, y^\star)$.

We remark that the assumption of strict NE in Theorem 1 cannot be weakened to the assumption of a unique, non-strict Nash equilibrium, as our empirical counterexample shows. Despite the isolated positive result given in Theorem 1 (under a very strong assumption that we do not expect to hold in practice), the negative empirical results encountered in this section paint a bleak picture of the last-iterate convergence of unmodified regret-matching algorithms. This sets the stage for the rest of the paper, where we will show unconditional last-iterate convergence of variants of $RM^+$.

## 4 CONVERGENCE PROPERTIES OF ExRM$^+$

In this section, we prove that ExRM$^+$ exhibits favorable last-iterate convergence results. The section is organized as follows. In Section 4.1, we prove asymptotic last-iterate convergence of ExRM$^+$. Then, in Section 4.2, we provide a concrete rate of $O(1/\sqrt{T})$ for the best iterate, based on which we finally show a linear last-iterate convergence rate using a restarting mechanism in Section 4.3. All omitted proofs for this section can be found in Appendix C.

### 4.1 ASYMPTOTIC LAST-ITERATE CONVERGENCE OF ExRM$^+$

ExRM$^+$ (Algorithm 4) is equivalent to the Extragradient (EG) algorithm of Korpelevich (1976) for solving a *variational inequality* $VI(\mathcal{Z}_{\geq}, F)$. For a closed convex set $\mathcal{S} \subseteq \mathbb{R}^n$ and an operator $G : \mathcal{S} \to \mathbb{R}^n$, the variational inequality problem $VI(\mathcal{S}, G)$ is to find $z \in \mathcal{S}$, such that $\langle G(z), z - z' \rangle \leq 0$ for all $z' \in \mathcal{S}$. We denote $SOL(\mathcal{S}, G)$ the solution set of $VI(\mathcal{S}, G)$. EG has been extensively studied in the literature, and its last-iterate convergence properties are known in several settings, including:

1. If $G$ is Lipschitz and *pseudo monotone with respect to the solution set* $SOL(\mathcal{S}, G)$, i.e., for any $z^\star \in SOL(\mathcal{S}, G)$, it holds that $\langle G(z), z - z^\star \rangle \geq 0$ for all $z \in \mathcal{S}$, then iterates produced by EG converge to a solution of $VI(\mathcal{S}, G)$ (Facchinei & Pang, 2003, Ch. 12);

2. If $G$ is Lipschitz and *monotone*, i.e., $\langle G(z) - G(z'), z - z' \rangle \geq 0$ for all $z, z' \in \mathcal{S}$, then iterates $\{z^t\}$ produced by EG have $O(\frac{1}{\sqrt{t}})$ last-iterate convergence such that $\langle G(z^t), z^t - z \rangle \leq O(\frac{1}{\sqrt{t}})$ for all $z \in \mathcal{S}$ (Golowich et al., 2020; Gorbunov et al., 2022a; Cai et al., 2022).

Unfortunately, these results do not apply directly to our case: although the operator $F$ (as defined in Equation (2)) is $L_F$-Lipschitz-continuous over $\mathcal{Z}_{\geq}$, it is not monotone or even pseudo monotone with respect to $SOL(\mathcal{Z}_{\geq}, F)$. However, we observe that $F$ satisfies the *Minty condition*: there exists a solution $z^\star \in SOL(\mathcal{Z}_{\geq}, F)$ such that $\langle F(z), z - z^\star \rangle \geq 0$ for all $z \in \mathcal{Z}_{\geq}$. The Minty condition is weaker than pseudo monotonicity with respect to $SOL(\mathcal{Z}_{\geq}, F)$ (note the different quantifiers $\forall$ and $\exists$ for $z^\star$ in the two conditions).

**Fact 1** ($F$ satisfies the Minty condition). *For any Nash equilibrium* $z^\star \in \Delta^{d_1} \times \Delta^{d_2}$ *of the matrix game $A$,* $\langle F(z), z - az^\star \rangle \geq 0$ *holds for all* $a \geq 1$.

*Proof.* Let $z^\star = (x^\star, y^\star) \in \Delta^{d_1} \times \Delta^{d_2}$ be a Nash equilibrium of the matrix game. For any $z = (Rx, Qy) \in \mathcal{Z}_\geq$, we have $\langle F(z), z - az^\star \rangle = -\langle F(z), az^\star \rangle = a(x^\top Ay^\star - (x^\star)^\top Ay) \geq 0$, using $\langle F(z), z \rangle = 0$ and the definition of Nash equilibrium. $\square$

We are not aware of last-iterate convergence results for variational inequalities under only the Minty condition, but with this condition and Lipschitzness, a standard analysis shows that the distance between any $z^\star$ that satisfies the Minty condition and the sequence $\{z^t\}$ produced by ExRM$^+$ is decreasing. This serves as an important cornerstone for the rest of our analysis.

**Lemma 1** (Adapted from Lemma 12.1.10 in (Facchinei & Pang, 2003))**.** *Let $z^\star \in \mathcal{Z}_\geq$ be a point such that $\langle F(z), z - z^\star \rangle \geq 0$ for all $z \in \mathcal{Z}_\geq$. Let $\{z^t\}$ be the sequence produced by ExRM$^+$. Then for every iteration $t \geq 0$ it holds that $\|z^{t+1} - z^\star\|^2 \leq \|z^t - z^\star\|^2 - (1 - \eta^2 L_F^2)\|z^{t+\frac{1}{2}} - z^t\|^2$.*

By Lemma 1, the sequence $\{z^t\}$ is bounded, so it has at least one limit point $\hat{z} \in \mathcal{Z}_\geq$. We next show that every limit point $\hat{z}$ lies in the solution set of $VI(\mathcal{Z}_\geq, F)$ and induces a Nash equilibrium of the matrix game $A$. Moreover, we have $\lim_{t \to \infty} \|z^t - z^{t+1}\| = 0$.

**Lemma 2.** *Let $\{z^t\}$ be the iterates produced by ExRM$^+$. Then $\lim_{t \to \infty} \|z^t - z^{t+1}\| = 0$ and also:*

1. *If $\hat{z}$ is a limit point of $\{z^t\}$, then $\hat{z} \in SOL(\mathcal{Z}_\geq, F)$.*

2. *If $\hat{z} \in SOL(\mathcal{Z}_\geq, F)$, then $(\hat{x}, \hat{y}) = g(\hat{z})$ is a Nash equilibrium of the matrix game $A$.*

Now we know that the sequence $\{z^t\}$ has at least one limit point $\hat{z} \in SOL(\mathcal{Z}_\geq, F)$. If $\hat{z}$ is the unique limit point, then $\{z^t\}$ converges to $\hat{z}$. To show this, we first provide another condition under which $\{z^t\}$ converges in the following proposition.

**Proposition 1.** *If the iterates $\{z^t\}$ produced by ExRM$^+$ have a limit point $\hat{z}$ such that $\hat{z} = az^\star$ for $z^\star \in \mathcal{Z}$ and $a \geq 1$ (equivalently, colinear with a pair of strategies), then $\{z^t\}$ converges to $\hat{z}$.*

*Proof.* Denote by $\{z_t\}_{t \in \kappa}$ a subsequence of $\{z^t\}$ that converges to $\hat{z}$. By Fact 1, the Minty condition holds for $\hat{z}$, so by Lemma 1, $\{\|z^t - \hat{z}\|\}$ is monotonically decreasing and therefore converges. Since $\lim_{t \to \infty} \|z^t - \hat{z}\| = \lim_{t(\in \kappa) \to \infty} \|z^t - \hat{z}\| = 0$, $\{z^t\}$ converges to $\hat{z}$. $\square$

However, the condition of Proposition 1 may not hold in general, and we empirically observe in experiments that it is not uncommon for the limit point $\hat{z} = (R\hat{x}, Q\hat{y})$ to have $R \neq Q$. To proceed, we will use the observation that the only "bad" case that prevents us from proving convergence of $\{z^t\}$ is that $\{z^t\}$ has infinitely-many limit points (note that the number of solutions $|SOL(\mathcal{Z}_\geq, F)|$ is indeed infinite). This is because if $\{z^t\}$ has a finite number of limit points, then since $\lim_{t \to \infty} \|z^{t+1} - z^t\| = 0$ (Lemma 2), it must have a unique limit point (see a formal proof in Proposition 3). In the following, to show that it is impossible that $\{z^t\}$ has infinitely many limit points, we first prove a lemma showing the structure of limit points of $\{z^t\}$.

**Lemma 3** (Structure of Limit Points)**.** *Let $\{z^t\}$ be the iterates produced by ExRM$^+$ and $z^\star \in \Delta^{d_1} \times \Delta^{d_2}$ be any Nash equilibrium of $A$. If $\hat{z}$ and $\tilde{z}$ are two limit points of $\{z^t\}$, then the following holds.*

1. *$\|az^\star - \hat{z}\|^2 = \|az^\star - \tilde{z}\|^2$ for all $a \geq 1$.*

2. *$\|\hat{z}\|^2 = \|\tilde{z}\|^2$.*

3. *$\langle z^\star, \hat{z} - \tilde{z} \rangle = 0$.*

See Figure 2 for an illustration of this lemma.[4] With such structural understanding of limit points, we are now ready to show that $\{z^t\}$ necessarily has a unique limit point.

Figure 2: Pictorial illustration of Lemma 3.

**Lemma 4** (Unique limit Point)**.** *The sequence $\{z^t\}$ produced by ExRM$^+$ has a unique limit point.*

---

[4]Note that we draw $z^\star$ in a simplex only as a simplified illustration—technically $z^\star$ should be from the Cartesian product of two simplexes instead.

**Proof Sketch** As discussed above, since $\lim_{t\to\infty}\|z^{t+1}-z^t\|=0$, it suffices to prove that $\{z^t\}$ has finitely many limit points. Let $\hat{z}=(\widehat{R}\hat{x},\widehat{Q}\hat{y})$ and $\tilde{z}=(\widetilde{R}\tilde{x},\widetilde{Q}\tilde{y})$ be any two distinct limit points of $\{z^t\}$ such that $\widehat{R}\neq\widehat{Q}$ and $\widetilde{R}\neq\widetilde{Q}$ (otherwise we can apply Proposition 1 to prove convergence). By careful case analysis, the structure of limit points (Lemma 3), and properties of Nash equilibrium, we prove a key equality: $\widehat{R}+\widetilde{R}=\widehat{Q}+\widetilde{Q}$. Now $\hat{z}$ and $\tilde{z}$ must be the only two limit points: suppose there exists another limit point $z=(Rx,Qy)$ with $R\neq Q$, then at least one of $\widehat{R}+R=\widehat{Q}+Q$ and $\widetilde{R}+R=\widetilde{Q}+Q$ would be violated and lead to a contradiction. Thus $\{z^t\}$ has at most two distinct limit points, which, again, when further combined with the fact $\lim_{t\to\infty}\|z^{t+1}-z^t\|=0$ implies that it in fact has a unique limit point (Proposition 3).

The last-iterate convergence of ExRM$^+$ now follows directly by Lemma 2 and Lemma 4.

**Theorem 2** (Last-Iterate Convergence of ExRM$^+$). *Let $\{z^t\}$ be the sequence produced by ExRM$^+$, then $\{z^t\}$ is bounded and converges to $z^\star\in\mathcal{Z}_\geq$ with $g(z^\star)=(x^\star,y^\star)\in\Delta^{d_1}\times\Delta^{d_2}$ being a Nash equilibrium of the matrix game $A$.*

### 4.2 Best-Iterate Convergence Rate of ExRM$^+$

To remedy not having a concrete convergence rate in the last result, we now prove an $O(\frac{1}{\sqrt{T}})$ best-iterate convergence rate of ExRM$^+$ in this section. The following key lemma relates the duality gap of a pair of strategies $(x^{t+1},y^{t+1})$ and the distance $\|z^{t+\frac{1}{2}}-z^t\|$.

**Lemma 5.** *Let $\{z^t\}$ be iterates produced by ExRM$^+$ and $(x^{t+1},y^{t+1})=g(z^{t+1})$. Then* $\text{DualityGap}(x^{t+1},y^{t+1})\leq\frac{12\|z^{t+\frac{1}{2}}-z^t\|}{\eta}$.

Now combining Lemma 1 and Lemma 5, we conclude the following best-iterate convergence rate.

**Theorem 3.** *Let $\{z^t\}$ be the sequence produced by ExRM$^+$ with initial point $z^0$. Then for any Nash equilibrium $z^\star$, any $T\geq 1$, there exists $t\leq T$ with $(x^t,y^t)=g(z^t)$ and*

$$\text{DualityGap}(x^t,y^t)\leq\frac{12\|z^0-z^\star\|}{\eta\sqrt{1-\eta^2 L_F^2}}\frac{1}{\sqrt{T}}.$$

*If $\eta=\frac{1}{\sqrt{2}L_F}$, then* $\text{DualityGap}(x^t,y^t)\leq\frac{24L_F\|z^0-z^\star\|}{\sqrt{T}}$.

*Proof.* Fix any Nash equilibrium $z^\star$ of the game. From Lemma 1, we know $\sum_{t=0}^{T-1}\left\|z^{t+\frac{1}{2}}-z^t\right\|^2\leq\frac{\|z^0-z^\star\|^2}{1-\eta^2 L_F^2}$. This implies that there exists $0\leq t\leq T-1$ such that $\|z^{t+\frac{1}{2}}-z^t\|\leq\frac{\|z^0-z^\star\|}{\sqrt{T}\sqrt{1-\eta^2 L_F^2}}$. We then get the desired result by applying Lemma 5. $\square$

### 4.3 Linear Last-Iterate Convergence for ExRM$^+$ with Restarts

In this section, based on the best-iterate convergence result from the last section, we further provide a simple restarting mechanism under which ExRM$^+$ enjoys linear last-iterate convergence. To show this, we recall that zero-sum matrix games satisfy the following *metric subregularity* condition.

**Proposition 2** (Metric Subregularity (Wei et al., 2021)). *Let $A\in\mathbb{R}^{d_1\times d_2}$ be a matrix game. There exists a constant $c>0$ (only depending on $A$) such that for any $z=(x,y)\in\Delta^{d_1}\times\Delta^{d_2}$, it holds that $\text{DualityGap}(x,y)\geq c\|z-\Pi_{\mathcal{Z}^\star}[z]\|$ where $\mathcal{Z}^\star$ denotes the set of all Nash equilibria.*

Together with the best-iterate convergence rate result from Theorem 3 (with $\eta=\frac{1}{\sqrt{2}L_F}$), we immediately get that for any $T\geq 1$, there exists $1\leq t\leq T$ such that

$$\left\|(x^t,y^t)-\Pi_{\mathcal{Z}^\star}[(x^t,y^t)]\right\|\leq\frac{\text{DualityGap}(x^t,y^t)}{c}\leq\frac{24L_F}{c\sqrt{T}}\cdot\left\|z^0-\Pi_{\mathcal{Z}^\star}[z^0]\right\|.$$

The above inequality further implies that if $T\geq\frac{48^2 L_F^2}{c^2}$, then there exists $1\leq t\leq T$ such that

$$\left\|(x^t,y^t)-\Pi_{\mathcal{Z}^\star}[(x^t,y^t)]\right\|\leq\frac{1}{2}\left\|z^0-\Pi_{\mathcal{Z}^\star}[z^0]\right\|.$$

Therefore, after at most a constant number of iterations (smaller than $\frac{48^2 L_F^2}{c^2}$), the distance of the best-iterate $(x^t, y^t)$ to the equilibrium set $\mathcal{Z}^\star$ is halved compared to that of the initial point. If we could somehow identify this best iterate, then we just need to restart the algorithm with this best iterate as the next initial strategy. Repeating this would then lead to a linear last-iterate convergence.

The issue in the argument above is obviously that we cannot exactly identify the best iterate since $\|(x^t, y^t) - \Pi_{\mathcal{Z}^\star}[(x^t, y^t)]\|$ is unknown. However, it turns out that we can use $\|z^{t+\frac{1}{2}} - z^t\|$ as a proxy since $\|(x^t, y^t) - \Pi_{\mathcal{Z}^\star}[(x^t, y^t)]\| \leq \frac{1}{c} \text{DualityGap}(x^t, y^t) \leq \frac{12\|z^{t+\frac{1}{2}} - z^t\|}{c\eta}$ by Lemma 5. This motivates the design of our algorithm: Restarting ExRM+ (RS-ExRM$^+$ for short; see Algorithm 6), which restarts for the $k$-th time if $\|z^{t+\frac{1}{2}} - z^t\|$ is less than $O(\frac{1}{2^k})$. Importantly, RS-ExRM$^+$ does not require knowing the value of $c$, the constant in the metric subregularity condition, which can be hard to compute.

| **Algorithm 6** Restarting ExRM$^+$ (RS-ExRM$^+$) | **Algorithm 7** Restarting SPRM$^+$ (RS-SPRM$^+$) |
|---|---|
| 1: **Input**: Step size $\eta \in (0, \frac{1}{L_F})$, $\rho > 0$. | 1: **Input**: Step size $\eta \in (0, \frac{1}{8L_F}]$. |
| 2: **Initialize**: $z^0 \in \mathcal{Z}$, $k = 1$ | 2: **Initialize**: $z^{-1} = w^0 \in \mathcal{Z}$, $k = 1$ |
| 3: **for** $t = 0, 1, \ldots$ **do** | 3: **for** $t = 0, 1, \ldots$ **do** |
| 4:     $z^{t+1/2} = \Pi_{\mathcal{Z}_\geq} (z^t - \eta F(z^t))$ | 4:     $z^t = \Pi_{\mathcal{Z}_\geq} (w^t - \eta F(z^{t-1}))$ |
| 5:     $z^{t+1} = \Pi_{\mathcal{Z}_\geq} (z^t - \eta F(z^{t+1/2}))$ | 5:     $w^{t+1} = \overline{\Pi}_{\mathcal{Z}_\geq} (w^t - \eta F(z^t))$ |
| 6:     **if** $\|z^{t+1/2} - z^t\| \leq \rho/2^k$ **then** | 6:     **if** $\|w^{t+1} - z^t\| + \|w^t - z^t\| \leq 8/2^k$ **then** |
| 7:        $z^{t+1} \leftarrow g(z^{t+1}) \in \Delta^{d_1} \times \Delta^{d_2}$ | 7:        $z^t, w^{t+1} \leftarrow g(w^{t+1}) \in \Delta^{d_1} \times \Delta^{d_2}$ |
| 8:        $k \leftarrow k + 1$ | 8:        $k \leftarrow k + 1$ |

The main result of this section is the following linear convergence rates of RS-ExRM$^+$.

**Theorem 4** (Linear Last-Iterate Convergence of RS-ExRM$^+$). *Let $\{z^t\}$ be the sequence produced by RS-ExRM$^+$ and let $\rho = \frac{4}{\sqrt{1-\eta^2 L_F^2}}$. Then for any $t \geq 1$, the iterate $(x^t, y^t) = g(z^t)$ satisfies*

$$\left\| (x^t, y^t) - \Pi_{\mathcal{Z}^\star}[(x^t, y^t)] \right\| \leq \frac{\text{DualityGap}(x^t, y^t)}{c} \leq \alpha \cdot (1 - \beta)^t$$

*where $\alpha = \frac{576}{c^2 \eta^2 (1 - \eta^2 L_F^2)}$ and $\beta = \frac{1}{3(1+\alpha)}$.*

**Proof sketch**    Let us denote by $t_k$ the iteration at which the $k$-th restart happens. According to the restart condition and Lemma 5, we know that at iteration $t_k$, the duality gap of $(x^{t_k}, y^{t_k})$ and its distance to $\mathcal{Z}^\star$ is at most $O(\frac{1}{2^k})$. For iterate $t \in [t_k, t_{k+1})$ at which the algorithm does not restart, we can use Theorem 3 to show that its performance is not much worse than that of $(x^{t_k}, y^{t_k})$. Then we prove $t_{k+1} - t_k$ is upper bounded by a constant for every $k$, which leads to a linear last-iterate convergence rate for all iterations $t \geq 1$.

## 5   LAST-ITERATE CONVERGENCE OF SMOOTH PREDICTIVE RM+

In this section we study another RM$^+$ variant, SPRM$^+$ (Algorithm 5). We present convergence results very similar to those in the last section for ExRM$^+$. Given the similarity and for the sake of conciseness, we only state the main results here, with all details and proofs deferred to Appendix D.

**Theorem 5** (Asymptotic Last-Iterate Convergence of SPRM$^+$). *Let $\{w^t\}$ and $\{z^t\}$ be the sequences produced by SPRM$^+$, then $\{w^t\}$ and $\{z^t\}$ are bounded and both converge to $z^\star \in \mathcal{Z}_\geq$ with $g(z^\star) = (x^\star, y^\star) \in \Delta^{d_1} \times \Delta^{d_2}$ being a Nash equilibrium of the matrix game $A$.*

**Theorem 6** (Best-Iterate Convergence Rate of SPRM$^+$). *Let $\{w^t\}$ and $\{z^t\}$ be the sequences produced by SPRM$^+$. For any Nash equilibrium $z^\star$ of the game, any $T \geq 1$, there exists $1 \leq t \leq T$ such that the iterate $g(w^t) \in \Delta^{d_1} \times \Delta^{d_2}$ satisfies*

$$\text{DualityGap}(g(w^t)) \leq \frac{10\|w^0 - z^\star\|}{\eta} \frac{1}{\sqrt{T}}.$$

*Moreover, for any Nash equilibrium $z^\star$ of the game, any $T \geq 1$, there exists $0 \leq t \leq T$ such that the iterate $g(z^t) \in \Delta^{d_1} \times \Delta^{d_2}$ satisfies*

$$\text{DualityGap}(g(z^t)) \leq \frac{18\|w^0 - z^\star\|}{\eta}\frac{1}{\sqrt{T}}.$$

We apply the same idea of restarting to SPRM$^+$ to design a new algorithm called Restarting SPRM$^+$ (RS-SPRM$^+$; see Algorithm 7) that has provable linear last-iterate convergence.

**Theorem 7** (Linear Last-Iterate Convergence of RS-SPRM$^+$). *Let $\{w^t\}$ and $\{z^t\}$ be the sequences produced by RS-SPRM$^+$. Define $\alpha = \frac{400}{c^2\eta^2}$ and $\beta = \frac{1}{3(1+\alpha)}$. Then for any $t \geq 1$, the iterate $g(w^t) \in \Delta^{d_1} \times \Delta^{d_2}$ satisfies*

$$\left\|g(w^t) - \Pi_{\mathcal{Z}^*}[g(w^t)]\right\| \leq \frac{\text{DualityGap}(g(w^t))}{c} \leq \alpha \cdot (1-\beta)^t.$$

*Moreover, the iterate $g(z^t) \in \Delta^{d_1} \times \Delta^{d_2}$ satisfies*

$$\left\|g(z^t) - \Pi_{\mathcal{Z}^*}[g(z^t)]\right\| \leq \frac{\text{DualityGap}(g(z^t))}{c} \leq 2\alpha \cdot (1-\beta)^t.$$

## 6 NUMERICAL EXPERIMENTS

Next, we numerically evaluate the last-iterate performance of each algorithm studied in this paper. We use the $3 \times 3$ matrix game instance from Section 3, the normal-form representations of Kuhn poker and Goofspiel, as well as 25 random matrix games of size $(d_1, d_2) = (10, 15)$ (for which we average the duality gaps across the instances and show the associated confidence intervals). More details on the games can be found in Appendix E. We set $\eta = 0.1$ for all algorithms with a stepsize and we initialize all algorithms at $((1/d_1)\mathbf{1}_{d_1}, (1/d_2)\mathbf{1}_{d_2})$. In every iteration, we plot the duality gap of $(x^t, y^t)$ for RM$^+$, PRM$^+$, and alternating PRM$^+$; the duality gap of $g(z^t)$ for ExRM$^+$ and RS-ExRM$^+$; the duality gap of $g(w^t)$ for SPRM$^+$ and RS-SPRM$^+$. The results are shown in Figure 3. For the $3 \times 3$ matrix game, we see that alternating PRM$^+$, ExRM$^+$, and SPRM$^+$ achieve machine precision after $10^3$ iterations (while others stay around $10^{-1}$ as discussed earlier). On Kuhn poker, PRM$^+$ and alternating PRM$^+$ have faster convergence before $10^3$ iterations, and perform on par with ExRM$^+$ and SPRM$^+$ after that. On Goofspiel, Alternating PRM$^+$ is again the best algorithm, although all algorithms (except RM$^+$) have comparable performance after $10^5$ iterations. Finally, on random instances, the last iterate performance of ExRM$^+$ and SPRM$^+$ vastly outperform RM$^+$, PRM$^+$, and alternating RM$^+$, but we note that alternating PRM$^+$ seems to outperform all other algorithms. Overall, these results are consistent with our theoretical findings of ExRM$^+$ and SPRM$^+$. That said, understanding the superiority of alternating PRM$^+$ (an algorithm that we do not analyze in this work and for which neither ergodic nor last-iterate convergence guarantees are known) remains open. We also numerically investigate the impact of restarting for RS-ExRM$^+$ and RS-SPRM$^+$. For the sake of space we provide our analysis in Appendix E, where we note that restarting does not significantly change the empirical convergence of ExRM$^+$ and SPRM$^+$, which is coherent with the fact that ExRM$^+$ and SPRM$^+$ (without restarting) already exhibit fast last-iterate convergence.

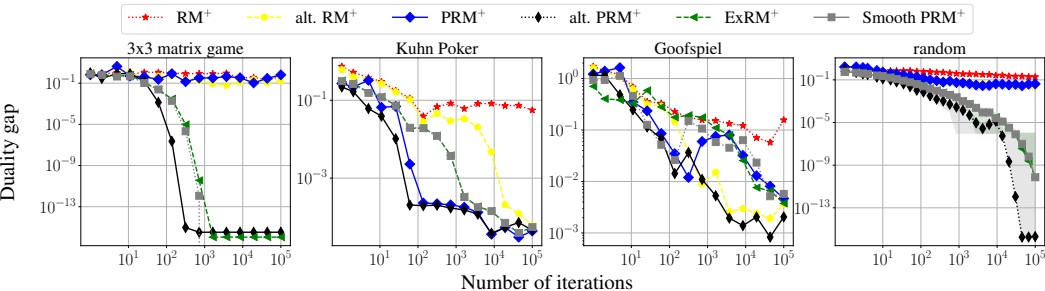

Figure 3: Empirical performances of several algorithms on the $3 \times 3$ matrix game (left plot), Kuhn poker and Goofspiel (center plots), and random instances (right plot).

## 7 CONCLUSIONS

In this paper, we investigate the last-iterate convergence properties of regret-matching algorithms, a class of extremely popular methods for equilibrium computation in games. Despite these methods enjoying strong *average* performance in practice, we show that unfortunately many practically-used variants might not converge to equilibrium in iterates. Motivated by these findings, we set out to investigate those variations with provable last-iterate convergence, establishing a suite of new results by using techniques from the literature on variational inequalities. For a restarted variant of these algorithms, we were able to prove, for the first time for regret matching algorithms, *linear-rate* convergence in iterates. Finally, we point out that several questions remain open, including giving concrete rates for the (non-restarted) ExRM$^+$, SPRM$^+$ and alternating PRM$^+$ algorithms.

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

## A  USEFUL PROPOSITIONS

**Proposition 3.** *Let $\mathcal{S} \subseteq \mathbb{R}^n$ be a compact convex set and $\{z^t \in \mathcal{S}\}$ be a sequence of points in $\mathcal{S}$ such that $\lim_{t \to \infty} \|z^t - z^{t+1}\| = 0$, then one of the following is true.*

1. *The sequence $\{z^t\}$ has infinite limit points.*

2. *The sequence $\{z^t\}$ has a unique limit point.*

*Proof.* We denote a closed Euclidean ball in $\mathbb{R}^n$ as $B(x, \delta) := \{x' \in \mathbb{R}^n : \|x' - x\| \leq \delta\}$. For the sake of contradiction, assume that neither of the two conditions are satisfied. This implies that the sequence $\{z^t\}$ has a finite number ($\geq 2$) of limit points. Then there exists $\epsilon > 0$ and two distinct limit points $\hat{z}_1$ and $\hat{z}_2$ such that

1. $\|\hat{z}_1 - \hat{z}_2\| \geq \epsilon$;

2. $\hat{z}_1$ is the only limit point of $\{z^t\}$ within $B(\hat{z}_1, \epsilon/2) \cap \mathcal{S}$;

3. $\hat{z}_2$ is the only limit point of $\{z^t\}$ within $B(\hat{z}_2, \epsilon/2) \cap \mathcal{S}$.

Given that $\lim_{t \to \infty} \|z^t - z^{t+1}\| = 0$, there exists $T > 0$ such that for any $t > T$, $\|z^t - z^{t+1}\| \leq \frac{\epsilon}{4}$. Now since $\hat{z}_1$ and $\hat{z}_2$ are limit points of $\{z^t\}$, there exists sub-sequences $\{z^{t_k}\}$ and $\{z^{t_j}\}$ that converges to $\hat{z}_1$ and $\hat{z}_2$, respectively. This means that there are infinite points from $\{z^{t_k}\}$ within $B(\hat{z}_1, \epsilon/4) \cap \mathcal{S}$ and infinite points from $\{z^{t_j}\}$ within $B(\hat{z}_2, \epsilon/4) \cap \mathcal{S}$. However, for $t > T$, there must also be infinite points from $\{z^t\}$ within the region $(B(\hat{z}_1, \epsilon/2) - B(\hat{z}_1, \epsilon/4)) \cap \mathcal{S}$. This implies there exists another limit point within $B(\hat{z}_1, \epsilon/2) \cap \mathcal{S}$ and contradicts the assertion that $\hat{z}_1$ is the only limit point within $B(\hat{z}_1, \epsilon/2) \cap \mathcal{S}$. $\qquad\square$

## B  PROOF OF THEOREM 1

*Proof of Theorem 1.* We will use an equivalent update rule of RM$^+$ (Algorithm 1) proposed by Farina et al. (2021): for all $t \geq 0$, $z^{t+1} = \Pi_{\mathbb{R}_{\geq 0}^{d_1 + d_2}}[z^t - \eta F(z^t)]$ where we use $z^t$ to denote $(R_x^t, R_y^t)$ and $F$ as defined in (2). The step size $\eta > 0$ only scales the $z^t$ vector and produces the same strategies $\{(x^t, y^t) \in \Delta^{d_1} \times \Delta^{d_2}\}$ as Algorithm 1. In the proof, we also use the following property of the operator $F$.

**Claim 1** (Adapted from Lemma 5.7 in (Farina et al., 2023)). *For a matrix game $A \in \mathbb{R}^{d_1 \times d_2}$, let $L_1 = \|A\|_{op} \sqrt{6 \max\{d_1, d_2\}}$ with $\|A\|_{op} := \sup\{\|Av\|_2 / \|v\|_2 : v \in \mathbb{R}^{d_2}, v \neq \mathbf{0}\}$. Then for any $z, z' \in \mathbb{R}_{\geq 0}^{d_1 + d_2}$, it holds that $\|F(z) - F(z')\| \leq L_1 \|g(z) - g(z')\|$.*

For any $t \geq 1$, since $z^{t+1} = \Pi_{\mathbb{R}_{\geq 0}^{d_1 + d_2}}[z^t - \eta F(z^t)]$, we have for any $z \in \mathbb{R}_{\geq 0}^{d_1 + d_2}$

$$\left\langle z^t - \eta F(z^t) - z^{t+1}, z - z^{t+1} \right\rangle \leq 0.$$
$$\Rightarrow \left\langle z^t - z^{t+1}, z - z^t \right\rangle \leq -\left\langle z^t - z^{t+1}, z^t - z^{t+1} \right\rangle + \left\langle \eta F(z^t), z - z^{t+1} \right\rangle. \tag{3}$$

By three-point identity, we have

$$\left\| z^{t+1} - z \right\|^2 = \left\| z^t - z \right\|^2 + \left\| z^{t+1} - z^t \right\|^2 + 2\left\langle z^t - z^{t+1}, z - z^t \right\rangle$$
$$\leq \left\| z^t - z \right\|^2 - \left\| z^{t+1} - z^t \right\|^2 + 2\left\langle \eta F(z^t), z - z^{t+1} \right\rangle. \tag{by (3)}$$

We define $k^t := \operatorname{argmin}_{k \geq 1} \|z^t - kz^\star\|$. With $z = k^t z^\star$, the above inequality is equivalent to

$$\|z^{t+1} - k^t z^\star\|^2 + 2\eta\langle F(z^\star), z^{t+1}\rangle$$
$$\leq \|z^t - k^t z^\star\|^2 + 2\eta\langle F(z^\star), z^t\rangle - \|z^{t+1} - z^t\|^2 + 2\eta\langle F(z^t) - F(z^\star), z^t - z^{t+1}\rangle$$
$$+ 2\eta\langle F(z^t), k^t z^\star\rangle \qquad \text{(We use } \langle F(z^t), z^t\rangle = 0)$$
$$\leq \|z^t - k^t z^\star\|^2 + 2\eta\langle F(z^\star), z^t\rangle + \eta^2\|F(z^t) - F(z^\star)\|^2 + 2\eta\langle F(z^t), k^t z^\star\rangle$$
$$\qquad\qquad (-a^2 + 2ab \leq b^2)$$
$$\leq \|z^t - k^t z^\star\|^2 + 2\eta\langle F(z^\star), z^t\rangle + \eta^2 L_1^2\|(x^t, y^t) - z^\star\|^2 + 2\eta\langle F(z^t), k^t z^\star\rangle.$$

Since $z^\star = (x^\star, y^\star)$ is a strict Nash equilibrium, by Lemma 6, there exists a constant $c > 0$ such that

$$2\eta\langle F(z^t), k^t z^\star\rangle = -2\eta k^t\big((x^t)^\top A y^\star - (x^\star)^\top A y^t\big)$$
$$\leq -2\eta c k^t\|(x^t, y^t) - z^\star\|^2$$
$$\leq -2\eta c\|(x^t, y^t) - z^\star\|^2,$$

where in the last inequality, we use $k^t \geq 1$ by definition. Thus, combining the above two inequalities gives

$$\|z^{t+1} - k^t z^\star\|^2 + 2\eta\langle F(z^\star), z^{t+1}\rangle \leq \|z^t - k^t z^\star\|^2 + 2\eta\langle F(z^\star), z^t\rangle + (\eta^2 L_1^2 - 2\eta c)\|(x^t, y^t) - z^\star\|^2.$$

Using definition of $k^{t+1}$ and choosing $\eta < \frac{c}{L_1^2}$, we have $2\eta c - \eta^2 L_1^2 > 0$ and

$$(2\eta c - \eta^2 L_1^2)\|(x^t, y^t) - z^\star\|^2 \leq \|z^t - k^t z^\star\|^2 + 2\eta\langle F(z^\star), z^t\rangle - \|z^{t+1} - k^{t+1} z^\star\|^2 - 2\eta\langle F(z^\star), z^{t+1}\rangle.$$

Telescoping the above inequality for $t \in [1, T]$ gives

$$(2\eta c - \eta^2 L_1^2)\sum_{t=1}^{T}\|(x^t, y^t) - z^\star\|^2 \leq \|z^1 - k^1 z^\star\|^2 + 2\eta\langle F(z^\star), z^1\rangle < +\infty$$

It implies that the sequence $\{(x^t, y^t)\}$ converges to $z^\star$. Moreover, it also gives a $O(\frac{1}{\sqrt{T}})$ best-iterate convergence rate with respect to $\|(x^t, y^t) - z^\star\|^2$. $\qquad\square$

**Lemma 6.** *If a matrix game $A$ has a strict Nash equilibrium $(x^\star, y^\star)$, then there exists a constant $c > 0$ such that for any $(x, y) \in \Delta^{d_1} \times \Delta^{d_2}$,*

$$x^\top A y^\star - (x^\star)^\top A y \geq c\|(x, y) - (x^\star, y^\star)\|^2.$$

*Proof.* Since $(x^\star, y^\star)$ is strict, it is also the unique and pure Nash equilibrium. we denote the unit vector $e_{i^\star} = x^\star$ with $i^\star \in [d_1]$[5] and constant $c_1 := \max_{i \neq i^\star} e_i^\top A y^\star - e_{i^\star}^\top A y^\star > 0$. Then, by definition, we have

$$x^\top A y^\star - (x^\star)^\top A y^\star = \sum_{i \neq i^\star} x[i]\big(e_i^\top A y^\star - e_{i^\star} A y^\star\big)$$
$$\geq c_1 \sum_{i \neq i^\star} x[i]$$
$$= \frac{c_1}{2}\sum_{i \neq i^\star} x[i] + \frac{c_1}{2}(1 - x[i^\star]) \qquad (\textstyle\sum_{i \in [d_1]} x[i] = 1)$$
$$\geq \frac{c_1}{2}\sum_{i \neq i^\star} x[i]^2 + \frac{c_1}{2}(1 - x[i^\star])^2 \qquad (x[i] \in [0, 1])$$
$$= \frac{c_1}{2}\|x - x^\star\|^2.$$

Similarly, denote $e_{j^\star} = x^\star$ with $j^\star \in [d_2]$ and constant $c_2 = (x^\star)^\top A y^\star - \max_{j \neq j^\star}(x^\star)^\top A e_j$, we get $(x^\star)^\top A y^\star - \max_{j \neq j^\star}(x^\star)^\top A y \geq \frac{c_2}{2}\|y - y^\star\|^2$. Combining the two inequalities above and let $c = \frac{1}{2}\min(c_1, c_2)$, we have $x^\top A y^\star - (x^\star)^\top A y \geq c\|(x, y) - (x^\star, y^\star)\|^2$ for all $(x, y)$. $\qquad\square$

---

[5]Given a positive integer $d$, we denote the set $\{1, 2, \ldots, d\}$ by $[d]$.

# C   PROOFS FOR SECTION 4

## C.1   PROOFS FOR SECTION 4.1

*Proof of Lemma 2.* From Lemma 1, we know

$$(1 - \eta^2 L_F^2) \sum_{t=0}^{\infty} \left\| z^{t+\frac{1}{2}} - z^t \right\|^2 \leq \left\| z^0 - z^\star \right\|^2.$$

Since $1 - \eta^2 L_F^2 > 0$, this implies

$$\lim_{t \to \infty} \left\| z^{t+\frac{1}{2}} - z^t \right\| = 0.$$

Using Proposition 4, we know that $\|z^{t+1} - z^{t+\frac{1}{2}}\| \leq \eta L_F \|z^{t+\frac{1}{2}} - z^t\|$. Thus

$$\lim_{t \to \infty} \left\| z^{t+1} - z^t \right\| \leq \lim_{t \to \infty} \left\| z^{t+1} - z^{t+\frac{1}{2}} \right\| + \left\| z^{t+\frac{1}{2}} - z^t \right\| = \lim_{t \to \infty} (1 + \eta L_F) \left\| z^{t+\frac{1}{2}} - z^t \right\| = 0.$$

If $\hat{z}$ is the limit point of the subsequence $\{z^t : t \in \kappa\}$, then we must have

$$\lim_{t(\in\kappa) \to \infty} z^{t+\frac{1}{2}} = \hat{z}.$$

By the definition of $z^{t+\frac{1}{2}}$ in Algorithm 4 and by the continuity of $F$ and of the projection, we get

$$\hat{z} = \lim_{t(\in\kappa) \to \infty} z^{t+\frac{1}{2}} = \lim_{t(\in\kappa) \to \infty} \Pi_{\mathcal{Z}_\geq} \left[ z^t - \eta F(z^t) \right] = \Pi_{\mathcal{Z}_\geq} [\hat{z} - \eta F(\hat{z})].$$

This shows that $\hat{z} \in SOL(\mathcal{Z}_\geq, F)$. Moreover, by $\hat{z} = \Pi_{\mathcal{Z}_\geq}[\hat{z} - \eta F(\hat{z})]$ and Lemma 7, we conclude $(\hat{x}, \hat{y}) = g(\hat{z}) \in \Delta^{d_1} \times \Delta^{d_2}$ is a Nash equilibrium of $A$. □

*Proof of Lemma 3.* Let $\{k_i \in \mathbb{Z}\}$ be an increasing sequence of indices such that $\{z^{k_i}\}$ converges to $\hat{z}$ and $\{l_i \in \mathbb{Z}\}$ be an increasing sequence of indices such that $\{z^{l_i}\}$ converges to $\tilde{z}$. Let $\sigma : \{k_i\} \to \{l_i\}$ be a mapping that always maps $k_i$ to a larger index in $\{l_i\}$, i.e. $\sigma(k_i) > k_i$. Such a mapping clearly exists. Since Lemma 1 applies to $az^\star$ for any $a \geq 1$, we get

$$\|az^\star - \hat{z}\|^2 = \lim_{i \to \infty} \left\| az^\star - z^{k_i} \right\|^2 \geq \lim_{i \to \infty} \left\| az^\star - z^{\sigma(k_i)} \right\|^2 = \|az^\star - \tilde{z}\|^2.$$

By symmetry, the other direction also holds. Thus $\|az^\star - \hat{z}\|^2 = \|az^\star - \tilde{z}\|^2$ for all $a \geq 1$.

Expanding $\|az^\star - \hat{z}\|^2 = \|az^\star - \tilde{z}\|^2$ gives

$$\|\hat{z}\|^2 - \|\tilde{z}\|^2 = 2a\langle z^\star, \hat{z} - \tilde{z}\rangle, \quad \forall a \geq 1.$$

It implies that $\|\hat{z}\|^2 = \|\tilde{z}\|^2$ and $\langle z^\star, \hat{z} - \tilde{z}\rangle = 0$. □

*Proof of Lemma 4.* For the sake of contradiction, let $\hat{z} = (\widehat{R}\hat{x}, \widehat{Q}\hat{y})$ and $\tilde{z} = (\widetilde{R}\tilde{x}, \widetilde{Q}\tilde{y})$ be any two distinct limit points of $\{z^t\}$. In the following, we will show a key equality:

**Claim 2.** $\widehat{R} + \widetilde{R} = \widehat{Q} + \widetilde{Q}$.

**Case 1:** $\hat{z} = (\widehat{R}x^\star, \widehat{Q}y^\star)$ **and** $\tilde{z} = (\widetilde{R}x^\star, \widetilde{Q}y^\star)$ **for a Nash equilibrium** $z^\star = (x^\star, y^\star)$. Part 2 of Lemma 3 gives

$$\|\hat{z}\|^2 = \|\tilde{z}\|^2 \Rightarrow (\widehat{R}^2 - \widetilde{R}^2)\|x^\star\|^2 = (\widetilde{Q}^2 - \widehat{Q}^2)\|y^\star\|^2.$$

Part 3 of Lemma 3 gives

$$\langle z^\star, \hat{z} - \tilde{z}\rangle = 0 \Rightarrow (\widehat{R} - \widetilde{R})\|x^\star\|^2 = (\widetilde{Q} - \widehat{Q})\|y^\star\|^2.$$

Combining the above two inequalities with the fact that $\hat{z} \neq \tilde{z}$, we get

$$\widehat{R} + \widetilde{R} = \widehat{Q} + \widetilde{Q}.$$

**Case 2:** $\hat{z} = (\widehat{R}\hat{x}, \widehat{Q}\hat{y})$ **and** $\tilde{z} = (\widetilde{R}\tilde{x}, \widetilde{Q}\tilde{y})$. Note that by Lemma 2, both $(\hat{x}, \hat{y})$ and $(\tilde{x}, \tilde{y})$ are Nash equilibrium of $A$ and thus $(\hat{x}, \tilde{y})$ is also a Nash equilibrium of $A$. Using Lemma 3, we have the following equalities

1.  $\|\hat{z}\|^2 = \|\tilde{z}\|^2 \Rightarrow \widehat{R}^2\|\hat{x}\|^2 - \widetilde{R}^2\|\tilde{x}\|^2 = \widetilde{Q}^2\|\tilde{y}\|^2 - \widehat{Q}^2\|\hat{y}\|^2$

2.  $0 = \langle(\hat{x}, \hat{y}), \hat{z} - \tilde{z}\rangle = \widehat{R}\|\hat{x}\|^2 - \widetilde{R}\langle\hat{x}, \tilde{x}\rangle + \widehat{Q}\|\hat{y}\|^2 - \widetilde{Q}\langle\hat{y}, \tilde{y}\rangle$

3.  $0 = \langle(\tilde{x}, \tilde{y}), \hat{z} - \tilde{z}\rangle = \widehat{R}\langle\hat{x}, \tilde{x}\rangle - \widetilde{R}\|\tilde{x}\|^2 + \widehat{Q}\langle\hat{y}, \tilde{y}\rangle - \widetilde{Q}\|\tilde{y}\|^2$

4.  $0 = \langle(\hat{x}, \tilde{y}), \hat{z} - \tilde{z}\rangle = \widehat{R}\|\hat{x}\|^2 - \widetilde{R}\langle\hat{x}, \tilde{x}\rangle + \widehat{Q}\langle\hat{y}, \tilde{y}\rangle - \widetilde{Q}\|\tilde{y}\|^2$

Combing the last three equalities gives

$$c := \widehat{R}\|\hat{x}\|^2 - \widetilde{R}\langle\hat{x}, \tilde{x}\rangle = \widehat{R}\langle\hat{x}, \tilde{x}\rangle - \widetilde{R}\|\tilde{x}\|^2,$$

and

$$-c = \widehat{Q}\|\hat{y}\|^2 - \widetilde{Q}\langle\hat{y}, \tilde{y}\rangle = \widehat{Q}\langle\hat{y}, \tilde{y}\rangle - \widetilde{Q}\|\tilde{y}\|^2.$$

Further combining the first equality gives

$(\widehat{R} + \widetilde{R})c + (\widehat{Q} + \widetilde{Q})(-c)$
$= \widehat{R}(\widehat{R}\|\hat{x}\|^2 - \widetilde{R}\langle\hat{x}, \tilde{x}\rangle) + \widetilde{R}(\widehat{R}\langle\hat{x}, \tilde{x}\rangle - \widetilde{R}\|\tilde{x}\|^2) + \widehat{Q}(\widehat{Q}\|\hat{y}\|^2 - \widetilde{Q}\langle\hat{y}, \tilde{y}\rangle) + \widetilde{Q}(\widehat{Q}\langle\hat{y}, \tilde{y}\rangle - \widetilde{Q}\|\tilde{y}\|^2)$
$= \widehat{R}^2\|\hat{x}\|^2 - \widetilde{R}^2\|\tilde{x}\|^2 - \widetilde{Q}^2\|\tilde{y}\|^2 + \widehat{Q}^2\|\hat{y}\|^2$
$= 0.$

If $c \neq 0$, then we get

$$\widehat{R} + \widetilde{R} = \widehat{Q} + \widetilde{Q}.$$

If $c = 0$, then $\widehat{R}\widetilde{R}\|\hat{x}\|^2\|\tilde{x}\|^2 = \widehat{R}\widetilde{R}\langle\hat{x}, \tilde{x}\rangle^2$ and $\widehat{Q}\widetilde{Q}\|\hat{y}\|^2\|\tilde{y}\|^2 = \widehat{Q}\widetilde{Q}\langle\hat{y}, \tilde{y}\rangle^2$ gives the equality condition of Cauchy-Schwarz inequality and we get $\hat{x} = \tilde{x}$ and $\hat{y} = \tilde{y}$. This reduced to Case 1 where we know $\widehat{R} + \widetilde{R} = \widehat{Q} + \widetilde{Q}$ holds. This completes the proof of the claim.

Note that $\widehat{R} + \widetilde{R} = \widehat{Q} + \widetilde{Q}$ must hold for any two distinct limit points $\hat{z}$ and $\tilde{z}$ of $\{z^t\}$. If $\widehat{R} = \widehat{Q}$, then $\hat{z} = \widehat{R}(\hat{x}, \hat{y})$ is colinear with a Nash equilibrium $(\hat{x}, \hat{y})$ and by Proposition 1, $\{z^t\}$ converges to $\hat{z}$ and $\hat{z}$ is the unique limit point. If $\widehat{R} \neq \widehat{Q}$, we argue that $\hat{z}$ and $\tilde{z}$ are the only two possible limit points. To see why, let us suppose that there exists another limit point $z = (Rx, Qy)$ distinct from $\hat{z}$ and $\tilde{z}$. If $R = \widehat{R}$, then by $R + \widetilde{R} = Q + \widetilde{Q}$, we know $Q = \widehat{Q}$. However, $R + \widehat{R} = 2\widehat{R} \neq 2\widehat{Q} = Q + \widehat{Q}$ gives a contradiction to Claim 2. If $R \neq \widehat{R}$, then combing

$$\widehat{R} + R = \widehat{Q} + Q, \quad \widetilde{R} + R = \widetilde{Q} + Q, \quad \widehat{R} + \widetilde{R} = \widehat{Q} + \widetilde{Q}$$

gives

$$\widehat{R} - \widehat{Q} = Q - R = \widetilde{R} - \widetilde{Q} = \widehat{Q} - \widehat{R}.$$

Thus, we have $\widehat{R} = \widehat{Q}$, and it contradicts the assumption that $\widehat{R} \neq \widehat{Q}$. Now we conclude that $\{z^t\}$ has at most two distinct limit points which means $\{z^t\}$ has a unique limit point given the fact that $\{z^t\}$ is bounded and $\lim_{t\to\infty}\|z^{t+1} - z^t\| = 0$. $\qquad\square$

### C.2 Proofs for Section 4.2

We will need the following lemma. Recall the normalization operator $g : \mathbb{R}_+^{d_1} \times \mathbb{R}_+^{d_2} \to \mathcal{Z}$ such that for $z = (z_1, z_2) \in \mathbb{R}_+^{d_1} \times \mathbb{R}_+^{d_2}$, we have $g(z) = (z_1/\|z_1\|_1, z_2/\|z_2\|_1) \in \mathcal{Z}$.

**Lemma 7.** *Let $z \in \Delta^{d_1} \times \Delta^{d_2}$ and $z_1, z_2, z_3 \in \mathcal{Z}_\geq$ such that $z_3 = \Pi_{\mathcal{Z}_\geq}[z_1 - \eta F(z_2)]$. Denote $(x_3, y_3) = g(z_3)$, then*

$$\text{DualityGap}(x_3, y_3) := \max_{y\in\Delta^{d_2}} x_3^\top Ay - \min_{x\in\Delta^{d_1}} x^\top Ay_3 \leq \frac{(\|z_1 - z_3\| + \eta L_F\|z_3 - z_2\|)(\|z_3 - z\| + 2)}{\eta}).$$

*Proof of Lemma 7.* Let $\hat{x} \in \text{argmin}_x \, x^\top A y_3$ and $\hat{y} \in \text{argmax}_y \, x_3^\top A y$. Then we have

$$
\begin{aligned}
&\max_y x_3^\top A y - \min_x x^\top A y_3 \\
&= x_3^\top A \hat{y} - \hat{x}^\top A y_3 \\
&= -\langle F(z_3), \hat{z} \rangle \\
&= \frac{1}{\eta}(\langle z_1 - z_3 + \eta F(z_3) - \eta F(z_2), z_3 - \hat{z} \rangle + \langle z_1 - \eta F(z_2) - z_3, \hat{z} - z_3 \rangle) \\
&\leq \frac{1}{\eta}\langle z_1 - z_3 + \eta F(z_3) - \eta F(z_2), z_3 - \hat{z} \rangle && (z_3 = \Pi_{\mathcal{Z}_\geq}[z_1 - \eta F(z_2)]) \\
&\leq \frac{(\|z_1 - z_3\| + \eta L_F \|z_3 - z_2\|)\|z_3 - \hat{z}\|}{\eta}.
\end{aligned}
$$

Moreover, we have

$$
\|z_3 - \hat{z}\| \leq \|z_3 - z\| + \|z - \hat{z}\| \leq \|z_3 - z\| + 2.
$$

Combining the above two inequalities completes the proof. $\qquad\square$

We now show a few useful properties of the iterates.

**Proposition 4.** *In the same setup of Lemma 1, for any $t \geq 1$, the following holds.*

1. $\|z^{t+1} - z^{t+\frac{1}{2}}\| \leq \eta L_F \|z^{t+\frac{1}{2}} - z^t\| \leq \|z^{t+\frac{1}{2}} - z^t\|$;

2. $\|z^{t+1} - z^t\| \leq (1 + \eta L_F)\|z^{t+\frac{1}{2}} - z^t\| \leq 2\|z^{t+\frac{1}{2}} - z^t\|$.

*Proof of Proposition 4.* Using the update rule of ExRM$^+$ and the non-expansiveness of the projection operator $\Pi_{\mathcal{Z}_\geq}[\cdot]$ we have

$$
\begin{aligned}
\left\| z^{t+1} - z^{t+\frac{1}{2}} \right\| &\leq \left\| z^t - \eta F(z^t) - (z^t - \eta F(z^{t+\frac{1}{2}})) \right\| \\
&= \eta \left\| F(z^{t+\frac{1}{2}}) - F(z^t) \right\| \leq \eta L_F \left\| z^{t+\frac{1}{2}} - z^t \right\|.
\end{aligned}
$$

Using triangle inequality, we have

$$
\left\| z^{t+1} - z^t \right\| \leq \left\| z^{t+1} - z^{t+\frac{1}{2}} \right\| + \left\| z^{t+\frac{1}{2}} - z^t \right\| \leq (1 + \eta L_F) \left\| z^{t+\frac{1}{2}} - z^t \right\|. \qquad\square
$$

We are now ready to prove Lemma 5.

*Proof of Lemma 5.* Let $z^\star$ be any Nash equilibrium of the game. Since $z^{t+1} = \Pi_{\mathcal{Z}_\geq}[z^t - \eta F(z^{t+\frac{1}{2}})]$, applying Lemma 7 gives

$$
\begin{aligned}
\text{DualityGap}(x^{t+1}, y^{t+1}) &\leq \frac{(\|z^{t+1} - z^t\| + \eta L_F\|z^{t+1} - z^{t+\frac{1}{2}}\|)(\|z^{t+1} - z^\star\| + 2)}{\eta} \\
&\leq \frac{3\|z^{t+\frac{1}{2}} - z^t\|(\|z^{t+1} - z^\star\| + 2)}{\eta} && (\eta L_F \leq 1 \text{ and Proposition 4}) \\
&\leq \frac{3\|z^{t+\frac{1}{2}} - z^t\|(\|z^0 - z^\star\| + 2)}{\eta} && (\text{Lemma 1}) \\
&\leq \frac{12\|z^{t+\frac{1}{2}} - z^t\|}{\eta}, && (z^0 \text{ and } z^\star \text{ are both in the simplex } \Delta^{d_1} \times \Delta^{d_2})
\end{aligned}
$$

which finishes the proof. $\qquad\square$

### C.3 Proofs for Section 4.3

*Proof of Theorem 4.* We note that RS-ExRM$^+$ always restarts in the first iteration since $\|z^{\frac{1}{2}} - z^0\| \leq \frac{\|z^0 - z^\star\|}{\sqrt{1 - \eta^2 L_F^2}} \leq \frac{2}{\sqrt{1 - \eta^2 L_F^2}}$. Let $1 = t_1 < t_2 < \ldots$ be the iterations where the algorithm restarts with $(x^{t_k}, y^{t_k})$. Using Lemma 5 and the restart condition, we know that for any $k \geq 1$,

$$\text{DualityGap}(x^{t_k}, y^{t_k}) \leq \frac{12\|z^{t_k - \frac{1}{2}} - z^{t_k - 1}\|}{\eta} \leq \frac{48}{\eta\sqrt{1 - \eta^2 L_F^2}} \cdot \frac{1}{2^k}$$

$$\left\| (x^{t_k}, y^{t_k}) - \Pi_{\mathcal{Z}^\star}[(x^{t_k}, y^{t_k})] \right\| \leq \frac{\text{DualityGap}(x^{t_k}, y^{t_k})}{c} \leq \frac{48}{c\eta\sqrt{1 - \eta^2 L_F^2}} \cdot \frac{1}{2^k}.$$

Moreover, for any iterate $t \in [t_k + 1, t_{k+1} - 1]$, using Lemma 5, we get

$$\text{DualityGap}(x^t, y^t) \leq \frac{12\|z^{t - \frac{1}{2}} - z^{t-1}\|}{\eta} \leq \frac{12\|z^{t_k} - \Pi_{\mathcal{Z}^\star}[(x^{t_k}, y^{t_k})]\|}{\eta\sqrt{1 - \eta^2 L_F^2}} \leq \frac{576}{c\eta^2(1 - \eta^2 L_F^2)} \cdot \frac{1}{2^k}.$$

Using metric subregularity (Proposition 2), we have for any iterate $t \in [t_k + 1, t_{k+1} - 1]$,

$$\left\| (x^t, y^t) - \Pi_{\mathcal{Z}^\star}[(x^t, y^t)] \right\| \leq \frac{576}{c^2\eta^2(1 - \eta^2 L_F^2)} \cdot \frac{1}{2^k}.$$

If we can prove $t_{k+1} - t_k$ is always bounded by a constant $C \geq 1$, then $t_k \leq C(k-1) + 1 \leq Ck$. Then for any $t \geq 1$, we have $t \geq t_{\lfloor \frac{t}{C} \rfloor}$ and

$$\left\| (x^t, y^t) - \Pi_{\mathcal{Z}^\star}[(x^t, y^t)] \right\| \leq \frac{576}{c^2\eta^2(1 - \eta^2 L_F^2)} \cdot \frac{1}{2^{\lfloor \frac{t}{C} \rfloor}} \leq \frac{576}{c^2\eta^2(1 - \eta^2 L_F^2)} \cdot 2^{-\frac{t}{C}} \leq \frac{576}{c^2\eta^2(1 - \eta^2 L_F^2)} \cdot (1 - \frac{1}{3C})^t.$$

**Bounding $t_{k+1} - t_k$** If $t_{k+1} - t_k = 1$, then the claim holds. Now we assume $t_{k+1} - t_k \geq 2$. Before restarting, the updates of $z^t$ for $t_k \leq t \leq t_{k+1} - 2$ is just ExRM$^+$. By Lemma 1, we know

$$\sum_{t = t_k}^{t_{k+1} - 2} \left\| z^{t + \frac{1}{2}} - z^t \right\|^2 \leq \frac{\|(x^{t_k}, y^{t_k}) - \Pi_{\mathcal{Z}^\star}[(x^{t_k}, y^{t_k})]\|^2}{1 - \eta^2 L_F^2}$$

It implies that there exists $t \in [t_k, t_{k+1} - 2]$ such that

$$\left\| z^{t + \frac{1}{2}} - z^t \right\| \leq \frac{\|(x^{t_k}, y^{t_k}) - \Pi_{\mathcal{Z}^\star}[(x^{t_k}, y^{t_k})]\|}{\sqrt{1 - \eta^2 L_F^2}\sqrt{t_{k+1} - t_k - 1}} \leq \frac{48}{c\eta(1 - \eta^2 L_F^2)\sqrt{t_{k+1} - t_k - 1} \cdot 2^k}.$$

On the other hand, since the algorithm does not restart in $[t_k, t_{k+1} - 2]$, we know for every $t \in [t_k, t_{k+1} - 2]$,

$$\left\| z^{t + \frac{1}{2}} - z^t \right\| > \frac{4}{\sqrt{1 - \eta^2 L_F^2} \cdot 2^{k+1}}.$$

Combining the above inequalities gives

$$t_{k+1} - t_k - 1 \leq \frac{48^2}{c^2\eta^2(1 - \eta^2 L_F^2)^2 \cdot 2^{2k}} \cdot \frac{(1 - \eta^2 L_F^2) \cdot 2^{2k+2}}{16}$$

$$= \frac{48^2/4}{c^2\eta^2(1 - \eta^2 L_F^2)} = \frac{576}{c^2\eta^2(1 - \eta^2 L_F^2)}.$$

**Linear Last-Iterate Convergence** Combing all the above, we get for all $t \geq 1$,

$$\text{DualityGap}(x^t, y^t) \leq \frac{576}{c\eta^2(1 - \eta^2 L^2)} \cdot \left( 1 - \frac{1}{3(1 + \frac{576}{c^2\eta^2(1 - \eta^2 L_F^2)})} \right)^t$$

$$\left\| (x^t, y^t) - \Pi_{\mathcal{Z}^\star}[(x^t, y^t)] \right\| \leq \frac{576}{c^2\eta^2(1 - \eta^2 L^2)} \cdot \left( 1 - \frac{1}{3(1 + \frac{576}{c^2\eta^2(1 - \eta^2 L_F^2)})} \right)^t.$$

This completes the proof. □

# D  PROOFS FOR SECTION 5

## D.1  ASYMPTOTIC CONVERGENCE OF SPRM$^+$

In the following, we will prove last-iterate convergence of SPRM$^+$ (Algorithm 5). The proof follows the same idea as that of ExRM$^+$ in previous sections. Applying standard analysis of OG, we have the following important lemma for SPRM$^+$.

**Lemma 8** (Adapted from Lemma 1 in (Wei et al., 2021)). *Let $z^\star \in \mathcal{Z}_\geq$ be a point such that $\langle F(z), z - z^\star \rangle \geq 0$ for all $z \in \mathcal{Z}_\geq$. Let $\{z^t\}$ and $\{w^t\}$ be the sequences produced by SPRM$^+$. Then for every iteration $t \geq 0$ it holds that*

$$\left\|w^{t+1} - z^\star\right\|^2 + \frac{1}{16}\left\|w^{t+1} - z^t\right\|^2 \leq \left\|w^t - z^\star\right\|^2 + \frac{1}{16}\left\|w^t - z^{t-1}\right\|^2 - \frac{15}{16}\left(\left\|w^{t+1} - z^t\right\|^2 + \left\|w^t - z^t\right\|^2\right).$$

*It also implies for any $t \geq 0$,*

$$\left\|w^{t+1} - z^t\right\| + \left\|w^t - z^t\right\| \leq 2\left\|w^0 - z^\star\right\|, \quad \left\|w^t - z^t\right\| + \left\|w^t - z^{t-1}\right\| \leq 2\left\|w^0 - z^\star\right\|.$$

**Lemma 9.** *Let $\{z^t\}$ and $\{w^t\}$ be the sequences produced by SPRM$^+$, then*

1. *The sequences $\{w^t\}$ and $\{z^t\}$ are bounded and thus have at least one limit point.*

2. *If the sequence $\{w^t\}$ converges to $\hat{w} \in \mathcal{Z}_\geq$, then the sequence $\{z^t\}$ also converges to $\hat{w}$.*

3. *If $\hat{w}$ is a limit point of $\{w^t\}$, then $\hat{w} \in SOL(\mathcal{Z}_\geq, F)$ and $g(\hat{w})$ is a Nash equilibrium of the game.*

*Proof.* By Lemma 8 and the fact that $w^0 = z^{-1}$, we know for any $t \geq 0$,

$$\left\|w^{t+1} - z^\star\right\|^2 + \frac{1}{16}\left\|w^{t+1} - z^t\right\|^2 \leq \left\|w^0 - z^\star\right\|^2,$$

which implies the sequences $\{w^t\}$ and $\{z^t\}$ are bounded. Thus, both of them have at least one limit point. By Lemma 8, we have

$$\frac{15}{32}\sum_{t=1}^T\left\|w^{t+1} - w^t\right\|^2 \leq \frac{15}{16}\sum_{t=1}^T\left(\left\|w^{t+1} - z^t\right\|^2 + \left\|w^t - z^t\right\|^2\right) \leq \left\|w^0 - z^\star\right\|^2 + \frac{1}{16}\left\|w^0 - z^{-1}\right\|^2.$$

This implies

$$\lim_{t\to\infty}\left\|w^{t+1} - w^t\right\| = 0, \quad \lim_{t\to\infty}\left\|w^t - z^t\right\| = 0, \quad \lim_{t\to\infty}\left\|w^{t+1} - z^t\right\| = 0.$$

If $\hat{w}$ is the limit point of the subsequence $\{w^t : t \in \kappa\}$, then we must have

$$\lim_{(t\in\kappa)\to\infty} w^{t+1} = \hat{w}, \quad \lim_{(t\in\kappa)\to\infty} z^t = \hat{w}.$$

By the definition of $w^{t+1}$ in SPRM$^+$ and by the continuity of $F$ and of the projection, we get

$$\hat{w} = \lim_{t(\in\kappa)\to\infty} w^{t+1} = \lim_{t(\in\kappa)\to\infty} \Pi_{\mathcal{Z}_\geq}\left[w^t - \eta F(z^t)\right] = \Pi_{\mathcal{Z}_\geq}[\hat{w} - \eta F(\hat{w})].$$

This shows that $\hat{w} \in SOL(\mathcal{Z}_\geq, F)$. By Lemma 7, we know $g(\hat{w})$ is a Nash equilibrium of the game. □

**Proposition 5.** *Let $\{z^t\}$ and $\{w^t\}$ be the sequences produced by SPRM$^+$. If $\{w^t\}$ has a limit point $\hat{w}$ such that $\hat{w} = az^\star$ for $z^\star \in \Delta^{d_1} \times \Delta^{d_2}$ and $a \geq 1$ (equivalently, colinear with a pair of strategies in the simplex), then $\{w^t\}$ converges to $\hat{w}$.*

*Proof.* By Fact 1, we know the MVI condition holds for $\hat{w}$, Then by Lemma 8, $\{\|w^t - \hat{w}\|^2 + \frac{1}{16}\|w^t - z^{t-1}\|^2\}$ is monotonitically decreasing and therefore converges. Let $\hat{w}$ be the limit of the sequence $\{w^t : t \in \kappa\}$. Using the fact that $\lim_{t\to\infty}\|w^t - z^{t-1}\| = 0$, we know that $\{\|w^t - \hat{w}\|\}$ also converges and

$$\lim_{t\to\infty}\left\|w^t - \hat{w}\right\|^2 + \frac{1}{16}\left\|w^t - z^{t-1}\right\|^2 = \lim_{t\to\infty}\left\|w^t - \hat{w}\right\|^2 = \lim_{t(\in\kappa)\to\infty}\left\|w^t - \hat{w}\right\|^2 = 0$$

Thus, $\{w^t\}$ converges to $\hat{w}$. □

**Lemma 10** (Structure of Limit Points). *Let $\{z^t\}$ and $\{w^t\}$ be the sequences produced by SPRM$^+$. Let $z^\star \in \Delta^{d_1} \times \Delta^{d_2}$ be any Nash equilibrium of A. If $\hat{w}$ and $\tilde{w}$ are two limit points of $\{w^t\}$, then the following holds.*

1. $\|az^\star - \hat{w}\|^2 = \|az^\star - \tilde{w}\|^2$ *for all* $a \geq 1$.

2. $\|\hat{w}\|^2 = \|\tilde{w}\|^2$.

3. $\langle z^\star, \hat{w} - \tilde{w}\rangle = 0$.

*Proof.* Let $\{k_i \in \mathbb{Z}\}$ be an increasing sequence of indices such that $\{w^{k_i}\}$ converges to $\hat{w}$ and $\{l_i \in \mathbb{Z}\}$ be an increasing sequence of indices such that $\{w^{l_i}\}$ converges to $\tilde{w}$. Let $\sigma : \{k_i\} \to \{l_i\}$ be a mapping that always maps $k_i$ to a larger index in $\{l_i\}$, i.e. $\sigma(k_i) > k_i$. Such a mapping clearly exists. Since Lemma 8 applies to $az^\star$ for any $a \geq 1$ and $\lim_{t\to\infty} \|w^t - z^{t-1}\| = 0$, we get

$$
\begin{aligned}
\|az^\star - \hat{w}\|^2 &= \lim_{i\to\infty} \left\|az^\star - w^{k_i}\right\|^2 + \frac{1}{16}\left\|w^{k_i} - z^{k_i-1}\right\|^2 \\
&\geq \lim_{i\to\infty} \left\|az^\star - w^{\sigma(k_i)}\right\|^2 + \frac{1}{16}\left\|w^{\sigma(k_i)} - z^{\sigma(k_i)-1}\right\|^2 \\
&= \|az^\star - \tilde{w}\|^2.
\end{aligned}
$$

By symmetry, the other direction also holds. Thus, $\|az^\star - \hat{w}\|^2 = \|az^\star - \tilde{w}\|^2$ for all $a \geq 1$.

Expanding $\|az^\star - \hat{w}\|^2 = \|az^\star - \tilde{w}\|^2$ gives

$$
\|\hat{w}\|^2 - \|\tilde{w}\|^2 = 2a\langle z^\star, \hat{w} - \tilde{w}\rangle, \quad \forall a \geq 1.
$$

It implies that $\|\hat{w}\|^2 = \|\tilde{w}\|^2$ and $\langle z^\star, \hat{w} - \tilde{w}\rangle = 0$. $\qquad\square$

With Lemma 8, Lemma 9, Proposition 5, and Lemma 10, by the same argument as the proof for ExRM$^+$ in Lemma 4, we can prove that $\{w^t\}$ produced by SPRM$^+$ has a unique limit point and thus converges, as shown in the following lemma.

**Lemma 11** (Uniqueness of Limit Point). *The iterates $\{w^t\}$ produced by SPRM$^+$ have a unique limit point.*

Thus, Theorem 5 directly follows from Lemma 9 and Lemma 11.

### D.2 BEST-ITERATE CONVERGENCE OF SPRM$^+$

**Lemma 12.** *Let $\{z^t\}$ and $\{w^t\}$ be the sequences produced by SPRM$^+$ (Algorithm 5). Then*

1. $\mathrm{DualityGap}(g(w^{t+1})) \leq \frac{5(\|w^t-z^t\|+\|w^{t+1}-z^t\|)}{\eta}$;

2. $\mathrm{DualityGap}(g(z^t)) \leq \frac{9(\|w^t-z^t\|+\|w^t-z^{t-1}\|)}{\eta}$.

.

*Proof of Lemma 12.* Since $w^{t+1} = \Pi_{\mathcal{Z}_\geq}[w^t - \eta F(z^t)]$, by Lemma 7, we know for any $z^\star \in \mathcal{Z}^\star$

$$
\begin{aligned}
\mathrm{DualityGap}(g(w^{t+1})) &\leq \frac{(\|w^{t+1}-w^t\| + \eta L_F\|w^{t+1}-z^t\|)(\|w^{t+1}-z^\star\|+2)}{\eta} \\
&\leq \frac{(1+\eta L_F)(\|w^t-z^t\|+\|w^{t+1}-z^t\|)(\|w^{t+1}-z^\star\|+2)}{\eta} \\
&\leq \frac{(1+\eta L_F)(\|w^t-z^t\|+\|w^{t+1}-z^t\|)(\|w^0-z^\star\|+2)}{\eta} \\
&\leq \frac{5(\|w^t-z^t\|+\|w^{t+1}-z^t\|)}{\eta},
\end{aligned}
$$

where in the second inequality we apply $\|w^{t+1} - w^t\| \le \|w^{t+1} - z^t\| + \|w^t - z^t\|$; in the third inequality we use $\|w^{t+1} - z^\star\| \le \|w^0 - z^\star\|$ by Lemma 8; in the last inequality we use $\eta L_F \le \frac{1}{8}$ and $\|w^0 - z^\star\| \le 2$ since $w^0, z^\star \in \Delta^{d_1} \times \Delta^{d_2}$.

Similarly, since $z^t = \Pi_{\mathcal{Z}_\ge}[w^t - \eta F(z^{t-1})]$, by Lemma 7, we know for any $z^\star \in \mathcal{Z}^\star$

$$
\begin{aligned}
&\mathrm{DualityGap}(g(z^t)) \\
&\le \frac{(\|z^t - w^t\| + \eta L_F \|z^t - z^{t-1}\|)(\|z^t - z^\star\| + 2)}{\eta} \\
&\le \frac{(1 + \eta L_F)(\|w^t - z^t\| + \|w^t - z^{t-1}\|)(\|w^t - z^\star\| + \|w^t - z^t\| + 2)}{\eta} \\
&\le \frac{(1 + \eta L_F)(\|w^t - z^t\| + \|w^t - z^{t-1}\|)(\|w^0 - z^\star\| + 2\|w^0 - z^\star\| + 2)}{\eta} \\
&\le \frac{9(\|w^t - z^t\| + \|w^t - z^{t-1}\|)}{\eta},
\end{aligned}
$$

where in the second inequality we apply $\|z^t - z^{t-1}\| \le \|z^t - w^t\| + \|w^t - z^{t-1}\|$ and $\|z^t - z^\star\| \le \|z^t - w^t\| + \|w^t - z^\star\|$; in the third inequality we use $\|w^{t+1} - z^\star\| \le \|w^0 - z^\star\|$ and $\|w^t - z^t\| \le 2\|w^0 - z^\star\|$ by Lemma 8; in the last inequality we use $\eta L_F \le \frac{1}{8}$ and $\|w^0 - z^\star\| \le 2$ since $w^0, z^\star \in \Delta^{d_1} \times \Delta^{d_2}$. $\qquad\square$

*Proof of Theorem 6.* Fix any Nash equilibrium $z^\star$ of the game. From Lemma 8, we know that

$$
\sum_{t=0}^{T-1} \left( \|w^{t+1} - z^t\|^2 + \|w^t - z^t\|^2 \right) \le \frac{16}{15} \cdot \|w^0 - z^\star\|^2.
$$

This implies that there exists $0 \le t \le T - 1$ such that $\|w^{t+1} - z^t\| + \|w^t - z^t\| \le \frac{2\|w^0 - z^\star\|}{\sqrt{T}}$ (we use $a + b \le \sqrt{2(a^2 + b^2)}$). By applying Lemma 12, we then get

$$
\mathrm{DualityGap}(g(w^{t+1})) \le \frac{5(\|w^t - z^t\| + \|w^{t+1} - z^t\|)}{\eta} \le \frac{10\|w^0 - z^\star\|}{\eta\sqrt{T}}.
$$

Similarly, using Lemma 8 and the fact that $w^0 = z^{-1}$, we know

$$
\sum_{t=0}^{T} \left( \|w^t - z^t\|^2 + \|w^t - z^{t-1}\|^2 \right) \le \sum_{t=0}^{T} \left( \|w^t - z^t\|^2 + \|w^{t+1} - z^t\|^2 \right) \le \frac{16}{15} \cdot \|w^0 - z^\star\|^2.
$$

This implies that there exists $0 \le t \le T$ such that $\|w^t - z^t\| + \|w^t - z^{t-1}\| \le \frac{2\|w^0 - z^\star\|}{\sqrt{T}}$ (we use $a + b \le \sqrt{2(a^2 + b^2)}$). By applying Lemma 12, we then get

$$
\mathrm{DualityGap}(g(z^t)) \le \frac{9(\|w^t - z^t\| + \|w^t - z^{t-1}\|)}{\eta} \le \frac{18\|w^0 - z^\star\|}{\eta\sqrt{T}}.
$$

$\qquad\square$

### D.3 Linear convergence of RS-SPRM$^+$.

*Proof of Theorem 7.* We note that Algorithm 7 always restarts in the first iteration $t = 1$ since $\|w^1 - z^0\| + \|w^0 - z^0\| \le \sqrt{\frac{32}{15}\|w^0 - z^\star\|^2} \le 4 = \frac{8}{2^1}$. We denote $1 = t_1 < t_2 < \ldots < t_k < \ldots$ the iteration where Algorithm 7 restarts for the $k$-th time, i.e., the "If" condition holds. Then by Lemma 12, we know for every $t_k$, $w^{t_k} = g(w^{t_k}) \in \mathcal{Z} = \Delta^{d_1} \times \Delta^{d_2}$ and

$$
\mathrm{DualityGap}(w^{t_k}) \le \frac{5(\|w^{t_k} - z^{t_k-1}\| + \|w^{t_k-1} - z^{t_k-1}\|)}{\eta} \le \frac{40}{\eta \cdot 2^k},
$$

and by metric subregularity (Proposition 2), we have

$$\left\|w^{t_k} - \Pi_{\mathcal{Z}^\star}[w^{t_k}]\right\| \leq \frac{\text{DualityGap}(g(w^{t_k}))}{c} \leq \frac{40}{c\eta \cdot 2^k}.$$

Then, for any iteration $t_k + 1 \leq t \leq t_{k+1} - 1$, using Lemma 8 and Lemma 12, we have

$$\text{DualityGap}(g(w^t)) \leq \frac{5(\|w^t - z^{t-1}\| + \|w^{t-1} - z^{t-1}\|)}{\eta} \qquad \text{(Lemma 12)}$$

$$\leq \frac{10\|w^{t_k} - \Pi_{\mathcal{Z}^\star}[w^{t_k}]\|}{\eta} \qquad \text{(Lemma 8)}$$

$$\leq \frac{400}{c\eta^2 \cdot 2^k}.$$

Then again by metric subregularity, we also get $\|g(w^t) - \Pi_{\mathcal{Z}^\star}[g(w^t)]\| \leq \frac{400}{c^2\eta^2 \cdot 2^k}$ for every $t \in [t_k + 1, t_{k+1} - 1]$.

Similarly, for any iteration $t_k \leq t \leq t_{k+1} - 1$, using Lemma 8 and Lemma 12, we have

$$\text{DualityGap}(g(z^t)) \leq \frac{9(\|w^t - z^t\| + \|w^t - z^{t-1}\|)}{\eta} \qquad \text{(Lemma 12)}$$

$$\leq \frac{18\|w^{t_k} - \Pi_{\mathcal{Z}^\star}[w^{t_k}]\|}{\eta} \qquad \text{(Lemma 8)}$$

$$\leq \frac{720}{c\eta^2 \cdot 2^k}.$$

Then by metric subregularity, we also get $\|g(z^t) - \Pi_{\mathcal{Z}^\star}[g(z^t)]\| \leq \frac{400}{c^2\eta^2 \cdot 2^k}$ for every $t \in [t_k, t_{k+1} - 1]$.

**Bounding $t_{k+1} - t_k$** Fix any $k \geq 1$. If $t_{k+1} = t_k + 1$, then we are good. Now we assume $t_{k+1} > t_k + 1$. By Lemma 8, we know

$$\sum_{t=t_k}^{t_{k+1}-2} \left(\left\|w^{t+1} - z^t\right\|^2 + \left\|w^t - z^t\right\|^2\right) \leq \frac{16}{15}\left\|w^{t_k} - \Pi_{\mathcal{Z}^\star}[w^{t_k}]\right\|^2.$$

This implies that there exists $t \in [t_k, t_{k+1} - 2]$ such that

$$\left\|w^{t+1} - z^t\right\| + \left\|w^t - z^t\right\| \leq \frac{2\|w^{t_k} - \Pi_{\mathcal{Z}^\star}[w^{t_k}]\|}{\sqrt{t_{k+1} - t_k - 1}} \leq \frac{80}{c\eta 2^k} \frac{1}{\sqrt{t_{k+1} - t_k - 1}}.$$

On the other hand, since Algorithm 7 does not restart in $[t_k, t_{k+1} - 2]$, we have for every $t \in [t_k, t_{k+1} - 2]$,

$$\left\|w^{t+1} - z^t\right\| + \left\|w^t - z^t\right\| > \frac{8}{2^{k+1}}.$$

Combining the above two inequalities, we get

$$t_{k+1} - t_k - 1 \leq \frac{400}{c^2\eta^2}.$$

**Linear Last-Iterate Convergence Rates** Define $C := 1 + \frac{400}{c^2\eta^2} \geq t_{k+1} - t_k$. Then $t_k \leq Ck$ and for any $t \geq 1$, we have $t \geq t_{\lfloor \frac{t}{C} \rfloor}$ and

$$\text{DualityGap}(g(w^t)) \leq \frac{400}{c\eta^2} \cdot 2^{-\lfloor \frac{t}{C} \rfloor} \leq \frac{400}{c\eta^2}\left(1 - \frac{1}{3C}\right)^t = \frac{400}{c\eta^2} \cdot \left(1 - \frac{1}{3(1 + \frac{400}{c^2\eta^2})}\right)^t.$$

Using metric subregularity, we get for any $t \geq 1$.

$$\left\|g(w^t) - \Pi_{\mathcal{Z}^\star}[g(w^t)]\right\| \leq \frac{400}{c^2\eta^2} \cdot \left(1 - \frac{1}{3(1 + \frac{400}{c^2\eta^2})}\right)^t.$$

Similarly, we have for any $t \geq 1$

$$\text{DualityGap}(g(z^t)) \leq \frac{720}{c\eta^2} \cdot \left(1 - \frac{1}{3(1 + \frac{400}{c^2\eta^2})}\right)^t, \left\|g(z^t) - \Pi_{\mathcal{Z}^\star}[g(z^t)]\right\| \leq \frac{720}{c^2\eta^2} \cdot \left(1 - \frac{1}{3(1 + \frac{400}{c^2\eta^2})}\right)^t.$$

This completes the proof. □

# E  ADDITIONAL NUMERICAL EXPERIMENTS

## E.1  GAME INSTANCES

Below we describe the extensive-form benchmark game instances we test in our experiments. These games are solved in their *normal-form* representation. For each game, we report the size of the payoff matrix in this representation.

**Kuhn poker**  Kuhn poker is a widely used benchmark game introduced by Kuhn (1950). At the beginning of the game, each player pays one chip to the pot, and each player is dealt a single private card from a deck containing three cards: jack, queen, and king. The first player can check or bet, i.e., putting an additional chip in the pot. Then, the second player can check or bet after the first player's check, or fold/call the first player's bet. After a bet of the second player, the first player still has to decide whether to fold or to call the bet. At the showdown, the player with the highest card who has not folded wins all the chips in the pot.

The payoff matrix for Kuhn poker has dimension $27 \times 64$ and 690 nonzeros.

**Goofspiel**  We use an imperfect-information variant of the standard benchmark game introduced by Ross (1971). We use a 4-rank variant, that is, each player has a hand of cards with values $\{1, 2, 3, 4\}$. A third stack of cards with values $\{1, 2, 3, 4\}$ (in order) is placed on the table. At each turn, a prize card is revealed, and each player privately chooses one of his/her cards to bid. The players do not reveal the cards that they have selected. Rather, they show their cards to a fair umpire, which determines which player has played the highest card and should receive the prize card as a result. In case of a tie, the prize is split evenly among the winners. After 4 turns, all the prizes have been dealt out and the game terminates. The payoff of each player is computed as the sum of the values of the cards they won.

The payoff matrix for Goofspiel has dimension $72 \times 7{,}808$ and $562{,}176$ nonzeros.

**Random instances.**  We consider random matrices of size $(10, 15)$. The coefficients of the matrices are normally distributed with mean 0 and variance 1 and are sampled using the `numpy.random.normal` function from the `numpy` Python package (Harris et al., 2020). In all figures, we average the last-iterate duality gaps over the 25 random matrix instances, and we also show the confidence intervals.

## E.2  EXPERIMENTS WITH RESTARTING

In this section, we compare the last-iterate convergence performances of ExRM$^+$, SPRM$^+$ and their restarting variants RS-ExRM$^+$ and RS-SPRM$^+$ as introduced in Section 4.3 and Section 5. We present our results when choosing a stepsize $\eta = 0.05$ in Figure 4.

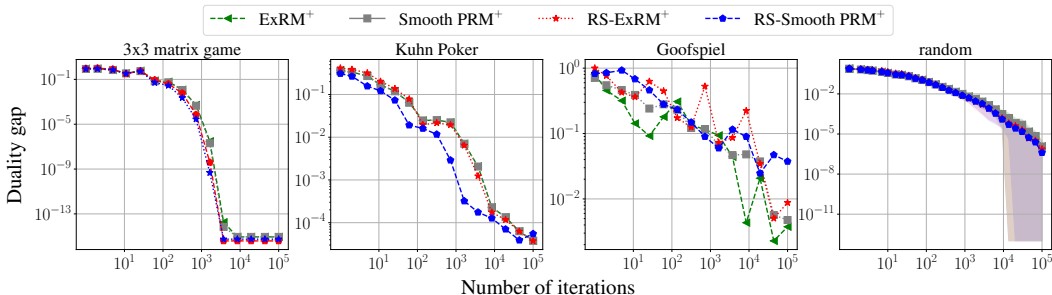

Figure 4: For a stepsize $\eta = 0.05$, empirical performances of several algorithms on our $3 \times 3$ matrix game (left plot), Kuhn Poker and Goofspiel (center plots) and on random instances (right plot).

