# OpenReview forum: "Last-Iterate Convergence Properties of Regret-Matching Algorithms in Games"
_ICLR.cc/2024/Conference — Submitted to ICLR 2024_

### Official Review · Reviewer_wbGe · 2023-10-21

**Soundness:** 4 excellent
**Presentation:** 4 excellent
**Contribution:** 4 excellent
**Rating:** 8
**Confidence:** 3

**Summary:**

This paper investigates the last-iterate convergence property of the algorithm family of Regret Matching, which is popular in practice but lacks corresponding theoretical guarantees for its good empirical performance. This paper fills the gap in this part. Specifically, the authors first show that RM+ and some of its variants, such as alternating RM+ and PRM+, may not converge by a toy example. Then, they prove that, under a very strong assumption of strict Nash equilibrium (NE), an assumption that usually does not hold in practice, RM+ enjoys last-iterate asymptotic convergence. Consequently, focusing on ExRM+, another algorithm in the RM+ family, the authors first prove that it enjoys last-iterate convergence when there is a unique limit point. Later, the authors rule out the case of infinite limit points and thus prove the last-iterate convergence of RM+. Besides, the authors prove an $O(1/\sqrt{T})$ best-iterate convergence rate for ExRM+. From the best-iterate analysis, they find a simple strategy for obtaining a linear last-iterate convergence rate is to restart the algorithm whenever the best-iterate comes. However, the distance to NE is not observable. To this end, they find a proxy variable (upper bound) for the distance to NE, restart the algorithm by checking the condition on the proxy variable, and eventually achieve a linear last-iterate convergence rate. Moreover, they also prove a similar linear rate for another RM+ variant called SPRM+, following the same flow as before. Finally, empirical studies validate the effectiveness of the proposed methods.

**Strengths:**

This is a solid paper from my point of view. RM+ algorithms do not draw enough attention in the literature because of the lack of theoretical guarantees. This paper fills a gap in this point. The presentation is quite clear and intuitive. It is worth noting that the authors also give some illustrating examples to help readers understand the paper better, which is very good. The preliminaries are sufficient for readers with little background knowledge in this field or about RM+ algorithms. The solution is described step by step, which is quite clear and intuitive. Although the final solution (restarting mechanism) is simple, the obtained results are important since they show that RM+ algorithms are not only useful in practice but also theoretically guaranteed.

**Weaknesses:**

I do not see major weaknesses in this paper.

**Questions:**

I do not have any questions since the presentation is clear and intuitive.

---

### Official Review · Reviewer_aUEP · 2023-10-27

**Soundness:** 3 good
**Presentation:** 3 good
**Contribution:** 3 good
**Rating:** 8
**Confidence:** 2

**Summary:**

The authors study the last-iterate and best-iterate convergence of RM+ and variants in in 2-player zero-sum games.
They show:
(1)	Empirically, RM+ and some of its simplest variants fail to converge on a specific 3X3 example recently used by Farina et al to show instability of RM+
(2)	Analytically, two slightly more elaborate variants of RM+, namely ExRM+ and SPRM+ do converge
(3)	Furthermore, restarting ExRM+ and SPRM+ at the right times improves the theoretical convergence rate to linear.

**Strengths:**

I think that the results are interesting. Last-iterate convergence to NE is an interesting question, and understanding which natural algorithms converge and which don’t is a natural question.

**Weaknesses:**

On the downside, to be honest I couldn’t find anything particularly innovative about the paper. Perhaps the careful analysis of ExRM+ in Section 4 would be interesting even to experts.

**Questions:**

1.	I’m confused about the notation \delta_{\ge}: aren’t all the strategies in \delta already probability distribution that whose sum is exactly 1?

2.	For the numerical run on the 3X3 algorithm, it would be very insightful to plot the actual trajectory of the algorithms (it should be relatively easy to plot in 2D since the strategies are probability distributions over 3 actions)

a.	Related, I’m curious if best-iterate convergence of RM+ etc is any better?

3.	The bulk of the work in Sections 4 and 5 is, AFAICT, to deal with solutions of the VI that are not NE. I’m a bit confused about it – when you introduced the VI notation I expected that NE would be the only solutions. Maybe you could give a simple example of a non-NE solution? That would really help build intuition.

4.	I probably missed something, but I didn’t understand why you need to talk about best iterate vs last iterate in Section 4.2. Does the guarantee of Lemma 1 not imply something about the convergence of last z^t?

5.	Appendix C, Claim 2, Case 2 typo: “equilibrium” should be “equilibria” (This is not a question :))

---

> ### Author Response · Authors · 2023-11-17
> **Response to Reviewer aUEP**
>
> Thank you for the very positive review. We address your questions below.
>
> **Q1**: *I’m confused about the notation \delta_{\ge}: aren’t all the strategies in \delta already probability distribution that whose sum is exactly 1?*
>
> **A**: It is correct that all the strategies in $\Delta$ are probability distributions whose sum is exactly 1. The ExRM+ and SPRM+ algorithm (proposed in [1]) use the regret operator (Equation 2) which is defined over the lifted space $\Delta_{\ge}:= \\{  u \in R+, 1^T u \ge 1 \\}$. We can map a point $z$ in $\Delta_{\ge}$ to a point in $\Delta$ by $\ell_1$ normalization: the normalized point $z’ = g(z) = z/||z||_1$ is a probability distribution.
>
> **Q2**: *For the numerical run on the 3X3 algorithm, it would be very insightful to plot the actual trajectory of the algorithms (it should be relatively easy to plot in 2D since the strategies are probability distributions over 3 actions)*
>
> **A**: Thanks for this suggestion. We have run additional experiments to obtain plots showing the trajectories of RM+, alt. RM+, PRM+, alt. PRM+,  ExRM+, and SPRM+ on the $3\times 3$ matrix. We plot the last 2000 iterations (out of $10^5$ iterations) of both players in 2D. The results show that the trajectories of RM+, alt. RM+, PRM+ cycle while alt. PRM+,  ExRM+, and SPRM+ converge. The empirical results can be found at this anonymized link: https://docdro.id/KUGvPvg
> We thank you for this interesting point, we will add this in the revised paper.
>
>
> **Q2a**. *Related, I’m curious if best-iterate convergence of RM+ etc is any better?*
>
> **A**:  No. As shown in Figure 1 in the paper, the duality gap of the best-iterate of RM+ is also at least $10^{-1}$ after $10^5$ iterates. Thus the non-convergence holds for both the best-iterate and last-iterate. Moreover, in additional experiments above, we observe the cycling behavior of RM+, which further shows that the best-iterate does not converge.
>
> **Q3**: *The bulk of the work in Sections 4 and 5 is, AFAICT, to deal with solutions of the VI that are not NE. I’m a bit confused about it – when you introduced the VI notation I expected that NE would be the only solutions. Maybe you could give a simple example of a non-NE solution? That would really help build intuition.*
>
> **A**: This is not correct. Every solution of the VI corresponds to a NE after normalization. In section 4 and 5, the main technical challenge is that some solutions do not satisfy the Minty condition. In the following, we first explain with an example that every solution of the VI corresponds to a NE after normalization.
>
> **Every solution of the VI corresponds to a NE after normalization**: Specifically, the VI is defined over the lifted space $Z_{\ge} = \Delta_{\ge}^{d_1} \times \Delta_{\ge}^{d_2}$  as defined above (also in page 4 at the paragraph introducing ExRM+ and SPRM+). Note that for  $x \in \Delta^{d_1}_{\ge}$, $\sum_i x_i$ could be greater than 1. Thus $x$ may not be a probability distribution and may not be in an NE. However, we have shown in Lemma 2 that for any solution $z = (x, y)$ of the VI in the lifted space (they may not be probability distributions), they corresponds to a Nash equilibrium in the sense that after $\ell_1$ normalization, the point $g(z) = (x/||x||_1, y /||y||_1)$ must be a Nash equilibrium.
>
> **Example**: The Rock-Paper-Scissors game has a unique Nash equilibrium where both players use $ x^* = y^*=  (\frac{1}{3}, \frac{1}{3}, \frac{1}{3})$. In the lifted space, any point of the form $(Qx^*, R y^*)$ with $ Q, R \ge 1$ is a solution to the variational inequality $VI(F, Z_{\ge})$. When $Q > 1$ or $R > 1$,  $z = (Qx^*, R y^*)$ is not a Nash equilibrium but corresponds to the Nash equilibrium $g(z) = (x^*, y^*)$ after $\ell_1$ normalization.
>
> **Some solutions do not satisfy the Minty condition**: Let $(x, y)$ be any Nash equilibrium. In Fact 1, we show that $(Q x, Qy)$ for any $Q \ge 1$ satisfies the Minty condition for the $VI(F, Z_\ge)$. However, there exists solutions of the form $(Qx, Ry)$ where $Q \ne R$ and such solutions may not satisfy the Minty condition. The existence of non-Minty solutions of the VI adds difficulty to proving last-iterate convergence. In section 4 and 5, we overcome this challenge by establishing structural characterization of the limit points of the learning dynamics (Lemma 3).
>
> **Q4**: *I probably missed something, but I didn’t understand why you need to talk about best iterate vs last iterate in Section 4.2. Does the guarantee of Lemma 1 not imply something about the convergence of last z^t?*
>
> A: Lemma 1 does not immediately imply last-iterate convergence, unless one first establishes stronger properties of the operator, which are not known to hold in general in our setting. The key is that Lemma 1 establishes geometric decrease in distance on the lifted joint strategy set $\mathcal{Z}_{\ge}$. However, this does not imply a geometric improvement of the distance for the normalized strategies.

---

> > ### Comment · Reviewer_aUEP · 2023-11-21
> >
> > Q2: The plots are very nice, but it would be helpful if you could e.g. color-code the trajectories over time so I can tell which iteration of x-player corresponds to y-player
> >
> > Q3: I didn't understand the point about Minty conditions. Could you provide an example?
> >
> > Q4: Could you provide an example where you have convergence in $Z_{\ge}$ but not in $Z$?
> >
> > Thanks for the discussion!

---

> > > ### Author Response · Authors · 2023-11-22
> > > **Response to Reviewer aUEP**
> > >
> > > Thanks for you comment! We address your questions below.
> > >
> > > **Q2**: *The plots are very nice, but it would be helpful if you could e.g. color-code the trajectories over time so I can tell which iteration of x-player corresponds to y-player*
> > >
> > > **A**: Thanks, we’ll do that!
> > >
> > > **Q3**: *I didn't understand the point about Minty conditions. Could you provide an example?*
> > >
> > > **A**: Sure! For a variational inequality $VI(Z, F)$, a point $z^* \in Z$ satisfies the Minty condition if $\langle F(z) , z - z^* \rangle \ge 0$ for all $z \in Z$. The following example shows that for the regret operator $F$ defined in equation (2), not all solutions in $SOL(Z, F)$ satisfy the Minty condition.
> > >
> > > **Example**: Let us consider the game with matrix $[[3,1], [4,5]]$ whose unique Nash equilibrium is $z^*=[x^* = [1, 0],  y^* = [1,0]]$. Consider any solution $z’ = (a x^*, b y^*)$ with $1 \le a < b/4$. Let $z = [x = [0, 1], y = [0,1]]$, then we can check $\langle F(z), z - z’ \rangle =  4a- b < 0$. Thus $z’$ does not satisfy the Minty condition.
> > >
> > > **Q4**: *Could you provide an example where you have convergence in $Z_{\ge}$ but not in $Z$?*
> > >
> > > **A**: There might be some miscommunication in the previous response. Our Theorem 2 proves last-iterate convergence in $Z_{\ge}$ which corresponds to a Nash equilibrium in $Z$. What we intended to convey is that Lemma 1 solely does not imply last-iterate convergence. To be specific, Lemma 1 shows that for a solution $z^*$ that satisfies the Minty condition, it holds that $||z^{t+1} - z^* ||^2 \le ||z^t - z^*||^2 - (1 - \eta^2 L^2) ||z^{t+1/2} - z^t||^2$ for all $t \ge 1$. However, this does not rule out the possibility of non-convergence of $\{z^t\}$. For example, the sequence $\{z^t\}$ may cycle around $z^*$, move closer to $z^*$ in each iteration but stay bounded away from $z^*$. We are not aware of any last-iterate convergence result that is solely based on Lemma 1 and does not require any other addtional assumptions. In the paper, to prove last-iterate convergence, we prove additional structural results of the limit points and then combine it with the interchangeability property of Nash equilibria in two-player zero-sum games.

---

### Official Review · Reviewer_RSe5 · 2023-10-30

**Soundness:** 3 good
**Presentation:** 3 good
**Contribution:** 2 fair
**Rating:** 5
**Confidence:** 3

**Summary:**

The research paper studies  the convergence properties of various variants of the Regret Matching algorithm within the context of two-player zero-sum games. The authors present a mix of favorable and unfavorable outcomes in this regard.

To begin with, the authors offer empirical evidence demonstrating that RM+, alternating RM+, and the more recently proposed Predictive RM+ (PRM+) do not admit last-iterate convergence even in the case of simple 3x3 zero-sum games. Subsequently, the authors introduce and evaluate two recently proposed variants of the RM algorithm.

Specifically they consider the Extragradient RM+ (ExRM+) and establish that it displays asymptotic last-iterate convergence and $O(1/\sqrt{T})$ best-iterate convergence. Furthermore, the authors demonstrate that a version of ExRM+ incorporating restarts achieves linear last-iterate convergence.
In section 5, the authors extend the above last-iterate results to Smooth Predicative RM+ (that is another variant of RM+). They respectively provide insights into the asymptotic last-iterate and $O(1/\sqrt{T})$ best-iterate convergence of SPRM+. Additionally, they extend their linear-convergence findings to the variant of SPRM+ that employs restarts.

**Strengths:**

Regret matching algorithms comprise an intriguing class of algorithms due to their practical success in large extensive form games. Despite the latter success, it is worth noting that the theoretical aspects of RM algorithms have not been sufficiently studied, as also acknowledged by the authors. From this perspective, I find the paper to be sufficiently motivated and to offer interesting insights into the behavior of these algorithms.

I am particularly intrigued by the observation that RM+, alt-RM+, and PRM+ do not exhibit last-iterate convergence, while the alternating version of PRM+ appears to outperform other RM variants in the experiments provided. This finding adds an interesting dimension to the discussion. Additionally, the paper is well-written and employs techniques that appear to distinguish from the previous techniques establishing last-iterate convergence for ExtraGradient and OMD (I have not yet delved into the appendix in full detail though).

**Weaknesses:**

Despite the fact that the paper provides some interesting insights on the last-iterate convergence properties of RM algorithms, I find that the paper lacks a key take-way message that creates me some doubts on the significance of the results.

As previously mentioned, the empirical success of RM+ algorithms lacks a solid theoretical foundation. In this context, investigating the last-iterate convergence properties of RM or RM+ appears to be a reasonable step in the right direction. However, I do have reservations about the significance of establishing last-iterate convergence results for artificial RM variants, especially given the existing results for Extragradient and OMD.

Additionally, I suggest that the paper could enhance its value by presenting time-average results for the different RM variants. It would be particularly intriguing to see the time-average convergence rates of the alternating PMR+ algorithm, which, as demonstrated in the provided experiments, seems to significantly outperform other RM methods in terms of last-iterate convergence. Furthermore, an experimental comparison of the last-iterate properties of Extragradient and OMD would be a valuable addition.

In summary, I consider this paper to be on the cusp of meeting the threshold for acceptance. It does contain some interesting findings, but I remain somewhat uncertain about their overall importance. I am open to reconsidering my assessment and potentially raising my score if the authors address the aforementioned comments during the rebuttal phase.

**Questions:**

Is there a limit for the constant $c$ in Proposition 2? Could it potentially be exceedingly small, perhaps even exponentially so?

---

> ### Author Response · Authors · 2023-11-17
> **Response to Reviewer RSe5**
>
> **Q**: *As previously mentioned, the empirical success of RM+ algorithms lacks a solid theoretical foundation. In this context, investigating the last-iterate convergence properties of RM or RM+ appears to be a reasonable step in the right direction. However, I do have reservations about the significance of establishing last-iterate convergence results for artificial RM variants, especially given the existing results for Extragradient and OMD.*
>
> **A:** We thank you for your time reviewing our work. In Figure 1 in the paper, we show that vanilla RM+, alternating RM+ and Predictive RM+ may not converge in iterates. This is the reason why we focus on Smooth Predictive RM+ (SPRM+) and ExRM+ [4].
> Additionally, we would like to clarify that our proofs of last-convergence for SPRM+ and ExRM+ do not follow from the existing analysis of Extragradient (EG) and optimistic mirror descent (OMD) [1]. Indeed, the results in [1] rely on the *motonicity* of the gradient operator. However, ExRM+ and SPRM+ are equivalent to running EG and OG using the **regret operator** $F(z)  := (Ay - x^T A y  \cdot 1_{d_1} , -A^Tx + x^T A y \cdot 1_{d_2})$ over a lifted space $Z_{\ge}$ (F is defined in equation (2) in the paper). The regret operator $F$ is **not monotone** over $Z_{\ge}$. That is why we cannot derive last-iterate convergence rates for ExRM+ or SPRM+ via existing results in [2]. The non-monotonicity of the regret operator also partly explains why the last-iterate convergence behavior of regret matching-type algorithms was previously not understood.
>
> **Q**: *Additionally, I suggest that the paper could enhance its value by presenting time-average results for the different RM variants. It would be particularly intriguing to see the time-average convergence rates of the alternating PRM+ algorithm, which, as demonstrated in the provided experiments, seems to significantly outperform other RM methods in terms of last-iterate convergence. Furthermore, an experimental comparison of the last-iterate properties of Extragradient and OMD would be a valuable addition.*
>
> **A**: We have conducted additional numerical experiments to answer your question.
> In the following anonymized link: https://docdro.id/6dvF63M, we plot in Figure 2 the average performance of SPRM+, ExRM+, alt PRM+, Extragradient algorithm (EG) and optimistic mirror descent (OMD). on the 3x3 matrix game from Section 3 in the paper. As evident from the plots, all these algorithms have similar empirical performances (for the average of the iterates).
> In the same document in Figure 1, we compare the last-iterate performance of all the algorithms presented in the paper, as well as EG and OMD. We see that ExRM+ and SPRM+ have similar performances as EG and OMD, and alt PRM+ is faster.
> We would like to conclude by highlighting two facts:
> 1. The performance of the average iterates of (predictive) RM+ with/without alternation have been studied in the past [3, 4].
> 2. While the alt PRM+ algorithm appears to have strong empirical performance, in the existing literature, there is no proof that alt PRM+ converges either on average or in iterates.
>
>
> We thank you for this interesting point, we will add this in the revised paper.
>
> **Q**: *Is there a limit for the constant $c$ in Proposition 2? Could it potentially be exceedingly small, perhaps even exponentially so?*
>
> **A**: The constant $c$ depends on the game matrix and can be arbitrarily small. The metric subregularity condition on the constant $c$ is introduced and utilized in previous works (see e.g., [2] and references therein). Our results are comparable to the linear last-iterate convergence results for OGDA [2], which also depend on the constant $c$. We remark that we also provide a $1/\sqrt{T}$ best-iterate convergence rate that is independent of $c$.
>
> [1] Finite-Time Last-Iterate Convergence for Learning in Multiplayer Games. Yang Cai, Argyris Oikonomou, Weiqiang Zheng. NeurIPS 2022
>
> [2] Wei, Chen-Yu, Chung-Wei Lee, Mengxiao Zhang, and Haipeng Luo. "Linear Last-iterate Convergence in Constrained Saddle-point Optimization." ICLR 2021
>
> [3] Burch, N., Moravcik, M., & Schmid, M.  Revisiting CFR+ and alternating updates. JAIR, 2019
>
> [4] Farina, Gabriele, Julien Grand-Clément, Christian Kroer, Chung-Wei Lee, and Haipeng Luo. "Regret Matching+:(In) Stability and Fast Convergence in Games. " NeurIPS, 2023

---

> > ### Comment · Reviewer_RSe5 · 2023-11-21
> > **Response to the authors**
> >
> > Thank you for your response and for taking the time to conduct and present the additional experiments. I understand that your techniques may differentiate from the previous last-iterate analysis which is definitely a merit of your work.
> >
> > The presented algorithm do not seem to appear any convergence advantage in the time-average sense while they admit similar performance with OMD and Extra-Gradient in the last-iterate sense (from the provided figures it seems that OMD and ExtraGradient admit comparable performance with alt PRM+ in the last-iterate sense).
> >
> > At the same time, the constant $c$ can be exponentially small with respect to the number of actions. As a result, the resulting dynamics may need exponential time (wrt to the number of actions) before reaching a min-max equilibrium. I understand though that the latter caveat appears also in previous works.
> >
> > To conclude I believe that your work contains some interesting results however I still reserve some doubts on the motivation.
> >
> > I will wait for your response in the AC's question regarding the novelty of your technical contribution before updating my score.

---

### Official Review · Reviewer_cEcU · 2023-11-03

**Soundness:** 2 fair
**Presentation:** 2 fair
**Contribution:** 2 fair
**Rating:** 3
**Confidence:** 4

**Summary:**

The paper considers the classical min-max matrix game and discusses the convergence properties of several algorithms.  In particular, the paper shows the last-iterate property holds for some algorithms, where the property means that the last update of the solution is the output of the algorithm. In general, many iterative algorithms based on online linear optimization (such as Hedge) have a solution based on the average of iterative updates. The experimental results compare several iterative algorithms.

**Strengths:**

The theoretical analyses are solid. In particular, the last-iterate property might be interesting for iterative algorithms, especially if the design is based on the online convex (linear) optimization. However, I feel that the asymptotic statement of the last-iterate property is not strong enough.

**Weaknesses:**

The min-max game itself is known to be an LP and thus solved in polynomial time. Iterative algorithms are one of the approaches to solving the LPs. The paper should consider comparing the state-of-the-art LP solvers with iterative algorithms. Although iterative algorithms are easy to implement, practical LP solvers have been improved for a long time and could be faster than naive iterative algorithms.

I do not understand why the last-iterate property is important. The convergence results can still be obtained by averaging the outputs of iterative OLO-based algorithms (such as Hedge). Maybe a more important issue is the speed itself, not whether the property holds or not.

I am afraid that the experimental data is rather too small for LP instances. For such small instances, I wonder that the sota LP solver such as Gurobi solve them much faster.

In summary, the paper focuses only on iterative algorithms, but as a solver of a certain LP, there are more alternatives to compare.

**Questions:**

Are the analyses for the last iterate property useful for constructing a new online-to-batch conversion technique (e.g., averaging all outputs of OCO algorithms)?

---

> ### Comment · Reviewer_aUEP · 2023-11-14
>
> I think the main point of the submission is to analyze plausible uncoupled dynamics by which players can converge to Nash equilibrium - not just an algorithm for computing one.
>
> This is why last iterate convergence is important, and why comparison to sota LP solvers is not so relevant.

---

> ### Comment · Reviewer_zjHV · 2023-11-14
>
> I am not sure I understand why the results of the authors is not something already known? Am I missing something?
>
> We already know that in convex-concave min-max games optimistic gradient descent ascent, and extragradient descent ascent converges in last-iterates. One of these algos (namely extragradient descent ascent) is one that the authors propose... but the convergence rate and last-iterate convergence is known (see for instance [1])?
>
> I am not disputing the correctness of the results of the paper, but I think that the authors' results are subsumed by previous papers.
>
> [1] Cai, Yang, Argyris Oikonomou, and Weiqiang Zheng. "Tight last-iterate convergence of the extragradient and the optimistic gradient descent-ascent algorithm for constrained monotone variational inequalities." arXiv preprint arXiv:2204.09228 (2022).

---

> > ### Author Response · Authors · 2023-11-15
> >
> > We thank you for your time reviewing the paper. Our results concerning the last-iterate convergence of ExRM+ and SPRM+ do not follow from the existing results in [1]. Indeed the authors in [1] focus on **monotone** operators, whereas in our setting the operator is **not** monotone. We provide more details below and we will make this more explicit in the revised version of our work.
> >
> > * A zero-sum game is a monotone game because its gradient operator $F_G(z)  := (Ay, -A^Tx)$ is a monotone operator (i.e., it satisfies $<F_G(z)- F_G(z’), z-z’> \ge 0$ for all $z, z’ \in \Delta^{d_1} \times \Delta^{d_2}$).
> >
> >
> > * However, the algorithms ExRM+ and SPRM+ are equivalent to running Extragradient and optimistic mirror descent using the **regret operator** $F(z)  := (Ay - x^T A y  \cdot 1_{d_1} , -A^Tx + x^T A y \cdot 1_{d_2})$ over a lifted space $Z_{\ge}$ (F is defined in equation (2) in the paper). The regret operator $F$ is **not monotone** over $Z_{\ge}$. That is why we can not derive last-iterate convergence rates for ExRM+ or SPRM+ by existing results in [1].
> >
> > [1] Cai, Yang, Argyris Oikonomou, and Weiqiang Zheng. "Tight last-iterate convergence of the extragradient and the optimistic gradient descent-ascent algorithm for constrained monotone variational inequalities." arXiv preprint arXiv:2204.09228 (2022).

---

> ### Author Response · Authors · 2023-11-17
> **Response to Reviewer cEcU: Part 1**
>
> **Q**: *The min-max game itself is known to be an LP and thus solved in polynomial time. Iterative algorithms are one of the approaches to solving the LPs. The paper should consider comparing the state-of-the-art LP solvers with iterative algorithms. Although iterative algorithms are easy to implement, practical LP solvers have been improved for a long time and could be faster than naive iterative algorithms.*
>
> **A**: Thanks for your comment! The focus on iterative algorithms like Regret Matching (RM), RM+ and their variants over Linear Programming (LP) methods is driven by two reasons.
>
> One of our motivations to study RM and its variants is that they are simple online learning algorithms, and understanding the convergence properties of uncoupled learning dynamics in games is an important question at the intersection of machine learning and game theory. In a multi-agent learning scenario, agents interact with each other and update their strategies in their own interest. Existence of simple learning dynamics that converge to Nash equilibrium in a day-to-day sense (i.e. in last iterate) provide stability guarantees for multi-agent systems, and provide some justification for using Nash equilibrium to predict the outcome of real-life multi-agent interactions. Prior to our work, the literature had focused on the behavior of optimistic variants of FTRL and OMD. Yet RM-type algorithms are very simple, and have a history of performing well in practice (more on that in the next paragraph), and thus their last-iterate convergence behavior is of interest in its own right.
>
> Our study is also motivated by the strong practical performance of RM-type iterative algorithms, especially in the context of large-scale extensive-form games. While LP methods might offer advantages in smaller games, their application becomes less feasible in larger, more complex scenarios. This is evident from the historical context: the last notable application of LP for offline two-player zero-sum games was in 2005, which successfully solved Rhode Island Hold’em [1, 2]. However, the practicality of LP diminishes as the game size increases. The size of LP formulations is linear in the game tree size, making the use of simplex or Interior Point Method (IPM) iterations extremely slow for larger games. Since 2007, there has been a significant shift in the field towards using iterative algorithms like Counterfactual Regret Minimization (CFR) [3], dilated Entropy-regularized algorithms such as excessive gap technique (EGT) and CFR+ [4]. We remark that CFR and CFR+ build upon RM and RM+, respectively. Every superhuman poker AI developed since then has utilized CFR+ [5, 6, 7]. Despite their strong practical performance in large-scale games, RM-type algorithms are not well-understood theoretically, which motivates our study.
>
> [1] Gilpin, Andrew, and Tuomas Sandholm. “Optimal Rhode Island Hold’em Poker.” AAAI, 2005
>
> [2] Gilpin, Andrew, and Tuomas Sandholm. "Lossless abstraction of imperfect information games." Journal of the ACM, 2007
>
> [3] Zinkevich, M., Johanson, M., Bowling, M., Piccione, C. Regret minimization in games with incomplete information. NeurIPS, 2007
>
> [4] Tammelin, Oskari. "Solving large imperfect information games using CFR+." 2014
>
> [5] Bowling, M., Burch, N., Johanson, M., & Tammelin, O. Heads-up limit hold’em poker is solved. Science, 2015
>
> [6] Brown, Noam, and Tuomas Sandholm. "Superhuman AI for heads-up no-limit poker: Libratus beats top professionals." Science, 2018
>
> [7] Brown, Noam, and Tuomas Sandholm. "Superhuman AI for multiplayer poker." Science, 2019

---

> ### Author Response · Authors · 2023-11-17
> **Response to Reviewer cEcU: Part 2**
>
> **Q**: *I do not understand why the last-iterate property is important. The convergence results can still be obtained by averaging the outputs of iterative OLO-based algorithms (such as Hedge). Maybe a more important issue is the speed itself, not whether the property holds or not.*
>
> **A**: We can average the iterates and get convergence results. However, last-iterate convergence is still important from both theoretical and practical point of view, as we have motivated in the introduction of the paper.
> If we view learning as a model of agents’ strategic behavior, then only the last-iterate convergence of learning dynamics guarantees that agents’ strategies (day-to-day behavior) stabilize at Nash equilibrium. Learning dynamics with only average-iterate convergence is unstable and exhibits cycling and divergence behavior. For example, the Hedge algorithm does not converge in last-iterate even for simple zero-sum games [8]. The problem of understanding last-iterate convergence for learning in games has received extensive attention in recent years, see e.g [9, 10, 11, 12, 13] and more references therein.
> Last-iterate is widely used in practice for simplicity. In some cases, iterate averaging can be cumbersome and even impractical when neural networks are involved. Thus it is crucial to develop theoretical tools to understand last-iterate convergence of learning algorithms.
>
>
> **Q**: *Are the analyses for the last iterate property useful for constructing a new online-to-batch conversion technique (e.g., averaging all outputs of OCO algorithms)?*
>
> **A**: We think the online-to-batch technique is not related to last-iterate convergence. This is because online convex optimization is a single-agent learning problem, where the goal is to minimize regret against an adversary; while learning in games is a multi-agent learning problem and the goal is to prove last-iterate convergence to Nash equilibrium. The following examples further illustrate the differences between regret minimization and last-iterate convergence in games.
>
> 1. The Extragradient (EG) algorithm suffers linear regret [11] in OCO but when both players employ EG in a zero-sum game, their joint strategy converges to a Nash equilibrium [13].
> 2. The online gradient descent (OGD) algorithm is a no-regret online algorithm but when both players employ OGD in a zero-sum game, their joint strategy diverges from Nash equilibrium [8].
>
> [8] Mertikopoulos, Panayotis, Christos Papadimitriou, and Georgios Piliouras. "Cycles in adversarial regularized learning." SODA, 2018
>
> [9] Daskalakis, Constantinos, and Ioannis Panageas. "Last-Iterate Convergence: Zero-Sum Games and Constrained Min-Max Optimization." ITCS, 2019
>
> [10] Lin, T., Zhou, Z., Mertikopoulos, P., & Jordan, M. I.. Finite-time last-iterate convergence for multi-agent learning in games. ICML 2020.
>
> [11] Golowich, N., Pattathil, S., & Daskalakis, C.  Tight last-iterate convergence rates for no-regret learning in multi-player games. NeurIPS, 2020
>
> [12] Wei, C. Y., Lee, C. W., Zhang, M., & Luo, H. Linear Last-iterate Convergence in Constrained Saddle-point Optimization. ICLR, 2021
>
> [13] Cai, Y., , Oikonomou, A., Zheng W. Finite-Time Last-Iterate Convergence for Learning in Multiplayer Games. NeurIPS 2022

---

### Official Review · Reviewer_zjHV · 2023-11-05

**Soundness:** 2 fair
**Presentation:** 4 excellent
**Contribution:** 2 fair
**Rating:** 6
**Confidence:** 3

**Summary:**

The authors study the last iterate convergence of regret matching algorithms in normal-form games. They show that a large class of well-known regret matching algorithms are not guaranteed to converge in last iterates and provide  EXRM+  and SPRM+ as alternatives that converge at a rate of 1/sqrt(T), and whose convergence rate can further improved to be linear

**Strengths:**

The non-convergence examples in last iterates are interesting and the linear convergence rates under restarting seem interesting.

**Weaknesses:**

The convergence results for EXRM+ and SPRM+ seem to be direct from known results (?) (see [1] for EXRM, and [2] for SPRM+). The authors claim that their setting does not satisfy the monotonocity assumption, but zero-sum bimatrix games do satisfy the monotonicity assumption (correct me if I am wrong?)

**Questions:**

Can you clarify the comments at the bottom of page 5? It is not clear to me why your setting is not a monotone game setting?

Can you explain why the restarted variants of your algorithms converge faster?

[1] Gorbunov, Eduard, Nicolas Loizou, and Gauthier Gidel. "Extragradient method: O (1/k) last-iterate convergence for monotone variational inequalities and connections with cocoercivity." International Conference on Artificial Intelligence and Statistics. PMLR, 2022.

[2] Cai, Yang, Argyris Oikonomou, and Weiqiang Zheng. "Tight last-iterate convergence of the extragradient and the optimistic gradient descent-ascent algorithm for constrained monotone variational inequalities." arXiv preprint arXiv:2204.09228 (2022).

---

> ### Author Response · Authors · 2023-11-14
>
> Thanks for your reivew and we address your questions below.
>
> **Q**: *Can you clarify the comments at the bottom of page 5? It is not clear to me why your setting is not a monotone game setting?*
>
> **A**: Our results do **not** follow from the referenced analysis of EG and OG [2]. We will clarify the misunderstanding below. First, monotonicity is defined on an operator. In the literature, a game is called monotone if its **gradient operator** is monotone: a zero-sum game is a monotone game because its gradient operator $F_G(z)  := (Ay, -A^Tx)$ is a monotone operator (i.e., it satisfies $\langle F_G(z)- F_G(z’), z-z’ \rangle \ge 0$ for all $z, z’ \in \Delta^{d_1} \times \Delta^{d_2}$). We can run EG and OG regardless of the monotonicity of the operator. The paper [2] shows that when the operator is monotone, EG and OG have last-iterate convergence rates, which is also acknowledged in the current paper (page 5, bullet point 2).
>
> However, the algorithms ExRM+ and SPRM+ (proposed by a recent paper [1]) studied in this paper are regret matching-type algorithms. ExRM+ and SPRM+ are equivalent to running EG and OG using the **regret operator** $F(z)  := (Ay - x^T A y  \cdot 1_{d_1} , -A^Tx + x^T A y \cdot 1_{d_2})$ over a lifted space $Z_{\ge}$ (F is defined in equation (2) in the paper). The regret operator $F$ is **not monotone** over $Z_{\ge}$. That is why we can not derive last-iterate convergence rates for ExRM+ or SPRM+ by existing results in [2]. The non-monotonicity of the regret operator also partly explains why last-iterate convergence of regret matching-type algorithms are poorly understood before.
>
> **Q**: *Can you explain why the restarted variants of your algorithms converge faster?*
>
> **A**: The short answer is that they may not converge faster. We prove sublinear $1/\sqrt{t}$ best-iterate convergence rates of ExRM+ and SPRM+ and linear last-iterate convergence of restarted variants of ExRM+ and SPRM+. But our experiments show that restarting does not significantly affects the last-iterate convergence speed of ExRM+ and SPRM+ (see Appendix E.2). Thus we conjecture that ExRM+ and SPRM+ without restarting already have linear last-iterate convergence. It would be very interesting to investigate this problem in the future.
>
> [1] Regret Matching+: (In)Stability and Fast Convergence in Games. Gabriele Farina, Julien Grand-Clément, Christian Kroer, Chung-Wei Lee, and Haipeng Luo. NeurIPS 2023
>
> [2] Finite-Time Last-Iterate Convergence for Learning in Multiplayer Games. Yang Cai, Argyris Oikonomou, Weiqiang Zheng. NeurIPS 2022

---

> > ### Comment · Reviewer_zjHV · 2023-11-21
> >
> > I thank the authors for their explanation, this is very helpful for me. The question that remains for me at this point to change my current score is: why should we care about about running EG and OG on the regret operator rather than the game gradient operator?

---

> > > ### Author Response · Authors · 2023-11-22
> > > **Response to Reviewer zjHV**
> > >
> > > Thanks for the discussion! We address you question below.
> > >
> > > **Q**: *I thank the authors for their explanation, this is very helpful for me. The question that remains for me at this point to change my current score is: why should we care about running EG and OG on the regret operator rather than the game gradient operator?*
> > >
> > > **A**: RM, and especially RM+, have been extremely important for large-scale game solving. In particular RM+ has been the regret minimizer of choice in every superhuman poker AI [1,2]. For this reason, RM+ is an important algorithm to understand in its own right. In spite of this, its last-iterate behavior is still mysterious. Although it was found empirically that the last strategy of RM+ is better than its average strategy [4], RM+ may diverge even on small matrix games as shown in our counterexample and in [3]. Understanding last-iterate properties of RM+ remains an important question in the extensive-form game community.
> > >
> > > To develop theoretical results for RM+ and its variants, it is important to understand the regret operator, which is non-monotone and not well-studied before. Our results for ExRM+ and SPRM+ proves last-iterate / best-iterate convergence with the non-monotone regret operator, which is technically interesting and shed light on further results on RM+ and its variants. EG and OG applied directly to the game gradient operator are important algorithms that achieve many theoretical results. However, these algorithms have not been competitive with RM+-based algorithms for large-scale games.
> > >
> > > [1] Brown, Noam, and Tuomas Sandholm. "Superhuman AI for heads-up no-limit poker: Libratus beats top professionals." Science, 2018
> > >
> > > [2] Brown, Noam, and Tuomas Sandholm. "Superhuman AI for multiplayer poker." Science, 2019
> > >
> > > [3] Lee, Chung-Wei, Christian Kroer, and Haipeng Luo. "Last-iterate convergence in extensive-form games." NeurIPS, 2021
> > >
> > > [4] Bowling, M., Burch, N., Johanson, M., & Tammelin, O. Heads-up limit hold’em poker is solved. Science, 2015

---

> > > > ### Comment · Reviewer_zjHV · 2023-11-23
> > > >
> > > > I thank the authors for their answer, I have increased my score accordingly to a 6. The explanation for my score is as follows:
> > > >
> > > > The paper studies an interesting problem and the counterexamples of RM+ not converging in last iterates, although known I don't think have been written down in a paper with spelled-out examples. That said, I am not totally convinced that there is a need to derive algorithms based on the regret operator rather than the game gradient operator. All this to say, my score should not invalidate the authors' contributions which are valuable but rather it reflects my relative ranking of the papers' contributions compared to other papers.

---

### Comment · Area_Chair_9L3a · 2023-11-21
**Discussion period closing soon - and a couple of questions**

Dear authors, dear reviewers,

As a reminder, the author-reviewer discussion period will be coming to close in about two days.

To make sure that this phase is as constructive as possible, I would kindly ask the reviewers, if you haven’t already done so, to go through the authors’ posted rebuttals, follow up on their replies to your comments, and engage with the authors if you would like to ask any further questions.

While on the topic of asking questions, let me ask a couple of my own:

1. In what way is Algorithm 1 (RM+) different from the classical algorithm of Hart and Mas-Colell (2000), who coined the term and introduced this class of algorithms 25 years ago? Is the algorithm of Tammelin (2014) different? If so, in what sense?

2. I am perplexed by Theorem 1. If I'm not mistaken, RM+ is simply projected descent on the operator $F$ defined in (2). Since $z^\ast$ is strict, $F$ satisfies the strict version of the Minty inequality of Fact 1, so the theorem would seem to follow from classical results in the VI literature - I'm thinking of the Facchinei-Pang textbook, but this result is likely more classical.

3. Lemma 12.1.10 of Facchinei-Pang indeed concerns pseudomonotone operators. However, from the very first line of its proof, it is clear that it only requires $x^{\ast}$ to be a solution of a Minty inequality. In view of this, how is Theorem 2 different from the standard result for extra-gradient applied to $F$?

4. A conceptual question: since the regret does not appear at all in ExRM+, in what sense is it a "regret-matching" algorithm?

Regards,

The  AC

---

> ### Author Response · Authors · 2023-11-21
> **Response to AC 9L3a**
>
> Thank you for your comment. We address you questions below.
>
> **Q1**: *In what way is Algorithm 1 (RM+) different from the classical algorithm of Hart and Mas-Colell (2000), who coined the term and introduced this class of algorithms 25 years ago? Is the algorithm of Tammelin (2014) different? If so, in what sense?*
>
> **A**: The classical algorithm of [1] is called Regret Matching (RM), while [2] proposes a variant of RM, called RM+. RM and RM+ are different and we provide their update rule below. The difference between RM and RM+ is that in each iteration, after observing a new utility, RM+ projects the cumulative regret $R^t$ onto the positive orthant, while RM does not perform such projections. As a result, the output strategy of RM+ is simply the $\ell_1$-normalization of $R^t$, while the output strategy of RM is to apply $\ell_1$-normalization only to positive coordinates of $R^t$.
>
> For the sake of clarity, we recall the update rule of RM and RM+ (for the $x$ player). Here we define $f(x, Ay) = Ay - x^T A y 1_{d_1}$ and $[ \cdot ]^+$ the projection operator onto the positive orthant.
>
> RM: $R^{t+1}_x = R^t_x - f(x^t, A y^t)$ and $x^{t+1} = [R^{t+1}]^+/ ||[R^{t+1}]^+ ||_1$;
>
> RM$^+$ :$R^{t+1}_x = [R^t_x - f(x^t, A y^t)]^+$ and $x^{t+1} = R^{t+1} / ||R^{t+1}||_1$.
>
> **Q2**: *I am perplexed by Theorem 1. If I'm not mistaken, RM+ is simply projected descent on the operator $F$ defined in (2). Since $z^*$ is strict, $F$ satisfies the strict version of the Minty inequality of Fact 1, so the theorem would seem to follow from classical results in the VI literature - I'm thinking of the Facchinei-Pang textbook, but this result is likely more classical.*
>
> **A**: For RM+, the regret operator $F$ is defined on the whole **positive orthant** $R_{+}^n$ and is not a Lipschitz operator on $R_{+}^n$ [3]. This is because we transform a point $z \in R_{+}^{n}$ back to $(x ,y)$ in the simplex to calculate the regret. Lack of Lipschitzness is the reason why Theorem 1 may not follow from classical results. We remark that for ExRM+ and SPRM+, the regret operator $F$ is restricted to the **clipped positive orthant**, which we denote as $\Delta_{\geq}$ in the paper, and $F$ is Lipschitz on $\Delta_{\geq}$.
>
> **Q3**: *Lemma 12.1.10 of Facchinei-Pang indeed concerns pseudomonotone operators. However, from the very first line of its proof, it is clear that it only requires $z^{\star}$ to be a solution to a Minty inequality. In view of this, how is Theorem 2 different from the standard result for extra-gradient applied to $F$?*
>
> **A**: This is a great question!  **Lemma 12.1.10** of [FP03] indeed only requires $z^*$ to be a solution to a Minty inequality and that is why in the paper we say Lemma 1 is adapted from Lemma 12.1.10 in [FP03]. However, the convergence of EG in **Theorem 12.1.11** [FP03] does require $F$ to be pseudomonotone with respect to the whole solution set (that is, for any $z^* \in SOL(Z, F)$,  it holds that  $\langle F(z), z -z^* \rangle \ge 0$ for all $z \in Z$). However, the regret operator $F$ defined in equation (2) does not satisfy this condition for the entire solution set (see an example below), and therefore Theorem 2 does not follow from Theorem 12.1.11 in [FP03]. To prove Theorem 2, we first characterize the limit points of the dynamics (Lemma 3) and then combine it with the interchangeability property of Nash equilibria in two-player zero-sum games.
>
> **Example**: Let us consider the game with matrix $[[3,1], [4,5]]$ whose unique Nash equilibrium is $z^*=[x^* = [1, 0],  y^* = [1,0]]$. Consider any solution $z’ = (a x^*, b y^*)$ with $1 \le a < b/4$. Let $z = [x = [0, 1], y = [0,1]]$, then we can check $\langle F(z), z - z’ \rangle =  4a- b < 0$. Thus $F$ is not pseudomonotone with respect to the whole solution set $SOL(\Delta_{\ge}, F)$.
>
> Thanks for mentioning this. We will add this discussion in the revised manuscript.
>
> **Q4**: *A conceptual question: since the regret does not appear at all in ExRM+, in what sense is it a "regret-matching" algorithm?*
>
> **A**: The operator $F$ defined in equation (2) is the regret operator, which is the same as RM and RM+. Specifically, the regret operator is defined as $F(z) = F((Rx, Qy)) = [f(x, Ay), f(y, -A^T x)] = [Ay - x^T Ay \cdot 1_{d_1}, -A^T x + x^T Ay \cdot 1_{d_2}]$ for all $z \in R_{+}^{d_1 + d_2}$.
>
> **References**
>
> [1] Hart, Sergiu, and Andreu Mas‐Colell. "A simple adaptive procedure leading to correlated equilibrium." Econometrica, 2000
>
> [2] Tammelin, Oskari. "Solving large imperfect information games using CFR+." 2014
>
> [3] Gabriele Farina, Julien Grand-Clément, Christian Kroer, Chung-Wei Lee, and Haipeng Luo. Regret Matching+: (In)Stability and Fast Convergence in Games. NeurIPS 2023
>
> [FP03] Facchinei, Francisco, and Jong-Shi Pang. Finite-dimensional variational inequalities and complementarity problems. 2003

---

> > ### Comment · Area_Chair_9L3a · 2023-11-21
> >
> > Thank you for your input, but my qustions still stand.
> >
> > 1. For the definition of RM+ versus RM: I understand the difference but you might then want to change the introduction (which puts the two in the same pot). This also raises a series of other questions: for instance, why RM+ and not RM (a classical and highly influential scheme)? Which properties of RM does RM+ inherit (since it's not the same algorithm, regret minimization is not automatic). Does the work of Tammelin and co-authors answer these questions? This is necessary context which is missing from your paper - and, apparently, is not clear to the reviewers either.
> >
> > 2. In your answer, you are saying that $F$ is not Lipschitz over the positive orthant. However, in your paper, right below Eq. (2), you are stating that $F$ is Lipschitz over the clipped positive orthant $\mathcal{Z}_{\geq}$, and you're even providing a value for its Lipschitz modulus with a reference to Farina et al. (2023) to that effect. Since the difference between the positive orthant and the clipped positive orthant is a corner-of-cube (over which $F$ is automatically Lipschitz since we're talking about a bounded set), this would imply that $F$ *is* Lipschitz. Claim 1 in Appendix B - itself adapted from the same paper by Farina et al. (2023) above - seems to confirm this. So, which is it?
> >
> > 3. Theorem 12.1.11 of Facchinei-Pang requires pseudomonotonicity only in order to invoke Lemma 12.1.10. So my question still stands as stated.
> >
> > 4. Since you are including a positive part in its definition, $R_t$ does not quite capture the definition of the regret - cf. part (1) of your own answer above. As such, I remain perplexed regarding the connection of ExtraRM+ with the players' regret.
> >
> > Regards,
> >
> > The AC

---

> > > ### Author Response · Authors · 2023-11-22
> > > **Response to AC 9L3a-Part 1**
> > >
> > > We address you questions below.
> > >
> > > **Q1**: *For the definition of RM+ versus RM: I understand the difference but you might then want to change the introduction (which puts the two in the same pot). This also raises a series of other questions: for instance, why RM+ and not RM (a classical and highly influential scheme)? Which properties of RM does RM+ inherit (since it's not the same algorithm, regret minimization is not automatic). Does the work of Tammelin and co-authors answer these questions? This is necessary context which is missing from your paper - and, apparently, is not clear to the reviewers either.*
> > >
> > > **A**: RM+ [2] is a well-known variant of RM [1], which has been used in some landmark AI breakthroughs, most notably in solving imperfect-information games such as poker. Both RM and RM+ are regret minimizers [1, 3]. In fact, as pointed out by [4], the difference between RM+ and RM fundamentally boils down to the difference between OMD and FTRL respectively. In the literature on extensive-form games, RM+ is typically considered the superior choice compared to RM, and that is why our paper focuses on it. Obtaining results for RM is also an interesting theoretical question, but the techniques we develop do not apply directly, and we prioritized RM+ over RM for the reason stated above. We will make sure to include more background on RM and RM+ and recall work establishing the practical superiority of RM+ over RM in the introduction.
> > >
> > > [1] Hart, Sergiu, and Andreu Mas‐Colell. "A simple adaptive procedure leading to correlated equilibrium." Econometrica, 2000
> > >
> > > [2] Tammelin, Oskari. "Solving large imperfect information games using CFR+." 2014
> > >
> > > [3] Tammelin, O., Burch, N., Johanson, M., & Bowling, M. Solving heads-up limit texas hold'em. IJCAI, 2015
> > >
> > > [4] Farina, Gabriele, Christian Kroer, Tuomas Sandholm. "Faster game solving via predictive blackwell approachability: Connecting regret matching and mirror descent." AAAI, 2021
> > >
> > > **Q2**: *In your answer, you are saying that $F$ is not Lipschitz over the positive orthant. However, in your paper, right below Eq. (2), you are stating that $F$ is Lipschitz over the clipped positive orthant $Z_{\ge}$, and you're even providing a value for its Lipschitz modulus with a reference to Farina et al. (2023) to that effect. Since the difference between the positive orthant and the clipped positive orthant is a corner-of-cube (over which
> > > $F$ is automatically Lipschitz since we're talking about a bounded set), this would imply that
> > > $F$ is Lipschitz. Claim 1 in Appendix B - itself adapted from the same paper by Farina et al. (2023) above - seems to confirm this. So, which is it?*
> > >
> > > **A**: The operator $F$ is not $L$-Lipschitz over the positive orthant for any $L > 0$. This fact can be seen as follows: take any point $z = (x, y)$ and $ z’ = (x’ , y’)$ in the simplex $\Delta^{d_1} \times \Delta^{d_2}$ such that $||F(z) - F(z’)|| = c$ where $c > 0$ is a constant. By the definition of $F$, $F(a z) = F(z)$ for any $a > 0$. Therefore, $||F(az) - F(az’)|| = ||F(z) - F(z’)|| = c$, but we can choose $a > 0$ small enough so that $||az - az’|| = a ||z - z’|| < c / L$. Then $||F(az) - F(az’)|| \le L ||az - az’|| < c$ gives a contradiction.
> > >
> > > Fundamentally, the reason why $F$ is not Lipschitz has to do with the fact that RM+ produces strategies of the form $R^t / || R^t||_1$, which is badly behaved when $R^t$ is close to the origin; this explains why Farina et al. (2023) modify the algorithms (ExRM+, SPRM+)  by “clipping” the origin away. We don’t understand your comment that $F$ should automatically be Lipschitz on any bounded set. This is definitely not an automatic conclusion, and it does not hold in our case for any bounded set that contains the origin.
> > >
> > > Regrettably, we had a typo (now fixed) in the statement of Claim 1 in the Appendix B (though we used the correct statement in Theorem 1). Claim 1 only claims a **weaker** Lipschitz-like inequality for $F$ over the **positive orthant**: $|| F(z) - F(z’)|| \le L ||g(z) - g(z’))||$ where $g$ is the $\ell_1$-normalization operator.  We remark that $g$ is Lipschitz over the **clipped positive orthant** (Proposition 1 of Farina et al. (2023)) and that’s why we get that $F$ is Lipschitz over the clipped positive orthant. Yet, $g$ is **not** Lipschitz when the origin is included—see discussion above.

---

> > > ### Author Response · Authors · 2023-11-22
> > > **Response to AC 9L3a-Part 2**
> > >
> > > **Q3**: *Theorem 12.1.11 of Facchinei-Pang requires pseudomonotonicity only in order to invoke Lemma 12.1.10. So my question still stands as stated.*
> > >
> > > **A**: Thanks for the expert question! Let us unpack the proof of Theorem 12.1.11 in [FP03] to see why it requires pseudo-monotonicity with respect to the whole solution set. In the first step, Theorem 12.1.11 shows that there is a limit point $\bar{x}$ of the sequence $\\{x^k\\}$ that belongs to the solution set $SOL(K, F)$. In the second step, it applies Lemma 12.1.10 with $x^* = \bar{x}$. However, this step requires that $\bar{x}$ satisfies the Minty condition in order to invoke Lemma 12.1.10, which may not hold unless *all* solutions in $SOL(K,F)$ satisfy the Minty condition.
> > >
> > > That is also why we prove additional structural results of the limit points to prove last-iterate convergence. We’ll expand on the discussion on this point in the paper.
> > >
> > > **Q4**: *Since you are including a positive part in its definition, $R^t$ does not quite capture the definition of the regret - cf. part (1) of your own answer above. As such, I remain perplexed regarding the connection of ExtraRM+ with the players' regret.*
> > >
> > > **A**:  Regarding the first part of your question on the definition of $R^t$ vs regret: We agree with you that RM+ does not maintain a regret vector; instead it maintains what you might call a cumulative positively thresholded regret vector. However, it is well-known that minimizing this thresholded regret vector implies that you also minimize regret (as shown by Tammelin et al [3], but see also [4]). RM+ is thus closely connected to the concept of regret as RM does: it is a regret minimizer and it maintains an upper bound of regret.
> > >
> > > Regarding the connection between RM+ and ExRM+: we see ExRM+ as the natural way to achieve an “extragradient-style” variant of RM+, since it uses the RM+ update rule. For that reason we think ExRM+ is a reasonable name, even if it is not a regret-minimizing algorithm.
> > >
> > > [3] Tammelin, O., Burch, N., Johanson, M., & Bowling, M. Solving heads-up limit texas hold'em. IJCAI, 2015
> > >
> > > [4] Farina, Gabriele, Christian Kroer, Tuomas Sandholm. "Faster game solving via predictive blackwell approachability: Connecting regret matching and mirror descent." AAAI, 2021

---

> > > > ### Comment · Area_Chair_9L3a · 2023-11-22
> > > >
> > > > Thank you for your continued input. In regard to your replies:
> > > > 1. If I'm not mistaken, the field of applications of RM+ concerns primarily extensive-form games, does it not? However, your paper concerns normal-form games, so my question stands: what is the concrete benefit of using RM+ over RM in a normal form game? And, if ExtraRM+ is not regret-minimizing, what is the benefit of using ExtraRM+ over "vanilla" extra-gradient gradient methods that *are* regret-minimizing?
> > > > 2. Regarding $F$: I am looking at the definition (2) of your paper - and, in particular, the second equality. Either there is (yet) another typo on the definition, or I don't understand how a function of the form $xAy$ is "badly behaved" at the origin.
> > > > 3. As long as the Minty inequality admits a solution, the proof of Theorem 12.1.11 of Facchinei-Pang goes through essentially verbatim (you can just take *that* solution of Minty instead of "an arbitrary" solution thereof).
> > > > 4. Your answer solidifies my concern: if RM+ is once-removed from RM, then ExtraRM+ is (at least) twice-removed, to the extent that I don't see any reasonable connection to regret-matching (though, to be fair, the reason for the disconnect here is RM+, not RM).
> > > >
> > > > At any rate, I do not want to monopolize the discussion further, so I am stopping here. Thank you again for your input.
> > > >
> > > > Regards,
> > > >
> > > > The AC

---

> > > > > ### Author Response · Authors · 2023-11-23
> > > > > **Response to AC 9L3a-Part 2**
> > > > >
> > > > > **Q3**: *As long as the Minty inequality admits a solution, the proof of Theorem 12.1.11 of Facchinei-Pang goes through essentially verbatim (you can just take that solution of Minty instead of "an arbitrary" solution thereof).*
> > > > >
> > > > > **A**: We disagree. Let us clarify this.
> > > > > 1. If only a subset $S$ of points of the solution set $SOL(K, F)$ satisfies the Minty condition, Lemma 12.1.10 does not hold. Although a modify version of it is true, i.e., for all points $x^* \in S$ that satisfy the Minty condition, the following inequality holds $||x^{t+1} - x^*||^2 \le ||x^t - x^*||^2 - (1 - \tau^2 L^2)||x^{t+1/2}  - x^t||^2$
> > > > > 2. Let us provide a sketch of the proof of Theorem 12.1.11, and explain where pseudomonotonicity (i.e., all the solutions satisfy the Minty condition) is needed. Proof Sketch: (1) F&P argues that the sequence $\\{x^k\\}$ is bounded, and therefore the sequence $\\{x^k\\}$ has at least one limit point $\bar{x}$; (2) the limit point $\bar{x}$ is in the solution set. Neither of these two steps requires pseudomonotonicity. However, psedudomonotonicity is indeed needed in the following key step.
> > > > > 3. (3) F&P invokes Lemma 12.1.10 to establish the following inequality w.r.t. $\bar{x}$: $$||x^{t+1} - \bar{x}||^2 \le ||x^t - \bar{x}||^2 - (1 - \tau^2 L^2)||x^{t+1/2}  - x^t||^2 (\*)$$ from which you can deduce that the sequence $\\{ ||x^k - \bar{x} || \\}$ is monotonically decreasing and hence the last-iterate convergence to $\bar{x}$.
> > > > > 4. However, $\bar{x}$ may not satisfy the Minty condition and the above inequality (\*) may not hold. In particular, the modified version of Lemma 12.1.10 does not imply inequality (*) for $\bar{x}$.
> > > > >
> > > > > **Q4**: *Your answer solidifies my concern: if RM+ is once-removed from RM, then ExtraRM+ is (at least) twice-removed, to the extent that I don't see any reasonable connection to regret-matching (though, to be fair, the reason for the disconnect here is RM+, not RM).*
> > > > >
> > > > > **A**: Whether the algorithms are close to RM should not be a concern since we have shown that RM+ is better than RM both theoretically and empirically. ExRM+, proposed in [FGK+23], is considered as a regret-matching type algorithm and has faster average-iterate convergence rate than RM / RM+. In this paper, we further provide best-iterate and last-iterate convergence results for ExRM+.
> > > > >
> > > > > [FGK+23] Gabriele Farina, Julien Grand-Clément, Christian Kroer, Chung-Wei Lee, and Haipeng Luo. Regret Matching+: (In)Stability and Fast Convergence in Games. NeurIPS 2023

---

> > > > > > ### Comment · Area_Chair_9L3a · 2023-11-23
> > > > > >
> > > > > > Dear authors,
> > > > > >
> > > > > > Thank you for unraveling the definition of $F$, I now (finally) understand your notation. However, I still don't understand the issue: since the operator is Lipschitz on the clipped orthant (by Farina et al.), and the sequence is eventually contained in any neighborhood of the clipped orthant (by Lemma 2), the lack of Lipschitz continuity at the origin seems inconsequential.
> > > > > >
> > > > > > As for Facchinei-Pang: if $z^\ast$ satisfies Minty, the proof of Theorem 12.1.10 immediately yields that any accumulation point $\bar z$ of $z_n$ is a solution point. [Incidentally, this is essentially the content of Lemma 2 in your paper, and the proof in the paper's appendix follows the same steps as FP (without a reference, though...)] Therefore, simply by unfolding the proofs of FP, we get that $z_n$ converges to the problem's solution set - and since the Nash equilibria of a min-max game are all indistinguishable in terms of payoffs, this means that the players' payoffs also converge.
> > > > > >
> > > > > > Am I wrong in the above? If not, since the theory of FP shows that $z_n$ converges to the problem's solution set and the players' payoffs converge to the value of the game, the main extra feature of Theorem 2 over FP seems to be the fact that $z_n$ actually converges to a solution *point* - as opposed to a set of solutions with identical payoffs. And this is where Lemmas 3 and 4 come in - do you concur?
> > > > > >
> > > > > > Regards,
> > > > > >
> > > > > > The AC

---

> > > > > > > ### Author Response · Authors · 2023-11-23
> > > > > > > **Response to Area Chair 9L3a**
> > > > > > >
> > > > > > > **Q**: *The lack of Lipschitz continuity at the origin seems inconsequential*
> > > > > > >
> > > > > > > **A**: The original question is about whether Theorem 1, which proves last-iterate convergence of RM+ when the game has a strict Nash equilibrium, is implied by classical results. As the AC acknowledged, the operator $F$ is not Lipschitz over the positive orthant so the results does not follow. If the argument of the AC was correct, then it would seemingly contradict the findings of Farina et al. (2023) that RM+ is indeed unstable, which could not be easily explained if Lipschitz continuity was inconsequential.
> > > > > > >
> > > > > > > If we understand correctly,  the **current** question is about ExRM+ (but on the positive orthant) and Theorem 2, and you propose the following to deal with the non-Lipschitzness of $F$ over the positive orthant:
> > > > > > > Start from a point $z^0$ far away from the origin, thus guaranteeing that the operator $F$ is Lipschitz-continuous close to the initial point
> > > > > > > Then, use Lemma 1 (the fact that $|| z^{t+1} - z^* || \le || z^t - z^* ||$, i.e, the distance to $z^*$ is decreasing) for a $z^*$ that satisfies the Minty condition
> > > > > > > Because of the decreasing condition, it would appear that the dynamics of $z^t$ are guaranteed to be sufficiently bounded away from the origin, thus escaping the ill behavior at or near the origin.
> > > > > > >
> > > > > > > The problem with the above argument is that it is impossible, in principle, to guarantee that $z^0$ and $z^*$ are closer to each other than $z^*$ to the origin. In other words, the decrease condition of Lemma 1 does not imply that the dynamics stay sufficiently bounded away from the origin, thus invalidating Lipschitness and further invalidating Lemmas 1 and 2 for the future, since Lemmas 1 & 2 requires Lipschitness at all times (and in particular, being away from the origin by some large enough amount).
> > > > > > >
> > > > > > > In fact, the idea that the AC suggested (running ExRM+ on the positive orthant without clipping) was also something that we discussed at length (through tweaks and variations to the different points above) and tried to make work in the months of work leading to this submission. Unfortunately, we found that this avenue was not viable. Any fix to the above argument would be 1) nontrivial and 2) would warrant a formal proof. It does **not** follow directly from what is proposed, and could very well be a different (and surprising) paper.
> > > > > > >
> > > > > > > ---
> > > > > > > **Q**: *On Theorem 12.1.11 of FP and its relationship to Lemma 2*
> > > > > > >
> > > > > > > **A**: Lemma 2 indeed builds on top of Theorem 12.1.11. We will add a pointer to that just like we did explicitly for Lemma 12.1.10 of FP in Lemma 1 (see Page 6).
> > > > > > >
> > > > > > > ---
> > > > > > > **Q**:  *On the main extra feature of Theorem 2 over FP*
> > > > > > >
> > > > > > > **A**: Your understanding is correct: our result for ExRM+ is convergence **in iterates** (as stated in Theorem 2), therefore implying convergence to the solution set and in payoffs. Note that convergence in iterates is highly desirable, as it implies that the learning dynamics stabilize. We believe that our techniques for establishing convergence in iterate without pseudomonotonicity are novel and possibly of independent interest.

---

> > > > > > > > ### Comment · Area_Chair_9L3a · 2023-11-23
> > > > > > > >
> > > > > > > > Thank you for your input. I remain perplexed regarding your choice of positioning relative to the classical VI literature but there is no need for further discussion at this point.
> > > > > > > >
> > > > > > > > Regards,
> > > > > > > >
> > > > > > > > The AC

---

> ### Author Response · Authors · 2023-11-23
> **Response to AC 9L3a-Part 1**
>
> **Q1**: *If I'm not mistaken, the field of applications of RM+ concerns primarily extensive-form games, does it not? However, your paper concerns normal-form games, so my question stands: what is the concrete benefit of using RM+ over RM in a normal form game? And, if ExtraRM+ is not regret-minimizing, what is the benefit of using ExtraRM+ over "vanilla" extra-gradient gradient methods that are regret-minimizing?*
>
> **A**: Here we provide one advantage of RM+ over RM.  RM corresponds to the lazy version of OGD while RM+ corresponds to the more standard agile version [FKS21]. While RM+ does not exactly "match regret", it in a sense matches something even better. For RM+, the maintained $R^t_{i}$ is not only an upper bound on the regret to action $i$ from round 1 to $t$, but also an upper bound on the regret from round $s$ to $t$ for any $s \le t$. In other words, RM+ provides a regret guarantee for any interval (the so-called adaptive regret guarantee by [HS07]). Our ultimate goal is to analyze RM+ for EFGs. But as the first step, we must understand the performance of RM+ for NFGs, a special case of EFGs. Note also that even in NFGs, RM+ performs better than RM (see e.g. [these lecture notes](http://www.columbia.edu/~ck2945/files/s20_8100/lecture_note_5_nash_from_rm.pdf), Figure 3). Providing further results for RM / RM+ in EFGs is an interesting future direction.
>
> We clarify that neither ExRM+ or extra-gradient (EG) is a regret minimization algorithm (See [GPD20 Appendix A.3 Proposition 10] for an example where EG suffers linear regret). But when both players run EG/ExRM+ in a two-player zero-sum game, each player’s regret vanishes and the dynamics converges to a Nash equilibrium. Moreover, we remark that the SPRM+ algorithm we analyze is a regret minimization algorithm.
>
> [HS07] Hazan, Elad, and Comandur Seshadhri. "Efficient learning algorithms for changing environments." ICML, 2007
>
> [FKS21] Farina, Gabriele, Christian Kroer, Tuomas Sandholm. "Faster game solving via predictive blackwell approachability: Connecting regret matching and mirror descent." AAAI, 2021
>
> [GPD20] Golowich, Noah, Sarath Pattathil, and Constantinos Daskalakis. "Tight last-iterate convergence rates for no-regret learning in multi-player games." NeurIPS, 2020
>
> **Q2**: *Regarding $F$: I am looking at the definition (2) of your paper - and, in particular, the second equality. Either there is (yet) another typo on the definition, or I don't understand how a function of the form $xAy$ is "badly behaved" at the origin.*
>
> **A**: There is no typo. Let us clarify again why $F$ is not Lipschitz over the positive orthant.
> 1. The definition of $F$ over the positive orthant $R_{+}^{d_1 + d_2}$ is as follows (the same as equation (2)): For a point $z \in R_{+}^{d_1 + d_2}$, we can always write it as $z = (Rx, Qy)$ where the $\ell_1$-norm $1^T x= 1^Ty = 1$ (so $(x, y) \in \Delta^{d_1} \times \Delta^{d_2}$ are strategies in the simplex). The operator $F$ is defined as
> $$F(z) = F((Rx, Qy)) = [ f(x, Ay), f(y, -A^T x) ] = [Ay - x^T A y 1_{d_1}, -A^T x + x^T A y 1_{d_2}].$$ Note that the argument of $F$ is not $(x,y)$, but $z$, so $xAy$ being nicely behaved does not imply $F$ being Lipschitz with respect to $z$.
> 2. Now it is clear that if two points $z, z’$ are very close to the origin, then $||z -z’||$ can be arbitrarily small. However, since the operator $F$ calculates regret by mapping $z = (Rx, Qy)$ back to $(x, y)$ in the simplex $\Delta^{d_1} \times \Delta^{d_2}$, the term $||F(z) - F(z’)||$ can be large. This is why $F$ is not Lipschitz over the positive orthant.
> 3. If we clip the origin and restrict $F$ on the clipped positive orthant $Z_{\ge}$, then $F$ is Lipschitz over the clipped positive orthant $Z_{\ge}$.

---

### Author Response · Authors · 2023-11-23
**End of Discussion Period**

With the author-reviewer discussion period coming to an end, we would like to thank again the AC and the reviewers for the time they invested and their work, which has helped improve our paper. Through the back-and-forth with the AC in particular, we hope to have convinced the reviewing team that
1. last-iterate convergence properties of RM+ and its variants are important;
2. our results are indeed interesting and nontrivial;
3. lack of Lipschitzness or pseudomonotonicity is a major obstacle when analyzing RM-type dynamics, and care must be applied when analyzing these dynamics as the theory landscape is rife with pitfalls rendering seemingly simple arguments much trickier than they appear.

---

### Meta-Review · Area_Chair_9L3a · 2023-12-06

**Metareview:**

This paper examines the convergence properties of certain "rergret-matching" algorithms - RM+ and its "extra-gradient" variant, ExtraRM+. The setting considered is that of finite two-player zero-sum games, and the authors provide a series of convergence results, of different types (last-iterate, best-iterate,...), both asymptotic and anytime.

This paper generated a very rich and extensive discussion at all stages of the reviewing process, and it oscillated between borderline positive and borderline negative assessments. In the end, even though the paper treats an interesting question, there were several concerns that remained:
- The paper's focus on RM+ / ExtraRM+ is not sufficiently well-motivated and justified. In their rebuttal, the authors pointed to the success of (Extra)RM+ in solving games in extensive form, but since the current paper concerns normal form games, this comparison is not fully convincing.
- Related to the above, even though the algorithms are called "regret-matching", they do not have the no-regret property, a fact which clashes with the existing literature on regret-matching algorithms, and makes the paper's contributions more difficult to position clearly in the context of related work.
- There were also concerns concerning the paper's technical contributions. In particular, given that the game's operator satisfies a Minty variational inequality, the lack of monotonicity is a minor issue in itself, and the lack of Lipschitzness was already treated in a recent paper by Farina et al. (2023). This would be less of a problem if the authors had explained clearly and concisely the tweaks required in the standard analysis of gradient / extra-gradient methods (see e.g., Facchinei & Pang, 2003), but the current presentation is somewhat unbalanced in this regard.

To sum up, even though this paper contains some interesting ideas and techniques, the current treatment falls short of being acceptable "as is" and a fresh round of reviews would be required before considering it again. For this reason, a decision was reached to make a "reject" recommendation while encouraging the authors to revise their paper from the ground up and resubmit.

**Justification For Why Not Higher Score:**

The paper has certain motivation and positioning issues which make the current version unsuitable for acceptance.

**Justification For Why Not Lower Score:**

N/A

---

### Decision · Program_Chairs · 2024-01-16

Reject